# The entorhinal cortex modulates trace fear memory formation and neuroplasticity in the mouse lateral amygdala via cholecystokinin

**Hemin Feng[1,2†], Junfeng Su[1,3‡], Wei Fang[1], Xi Chen[1,3], Jufang He[1,3*]**

[1]Departments of Neuroscience and Biomedical Sciences, City University of Hong Kong, Hong Kong, China; [2]Centre for Regenerative Medicine and Health, Hong Kong Institute of Science & Innovation, Chinese Academy of Sciences, Hong Kong SAR, China; [3]City University of Hong Kong Shenzhen Research Institute, Shenzhen, Guangzhou, China

**Abstract** Although fear memory formation is essential for survival and fear-related mental disorders, the neural circuitry and mechanism are incompletely understood. Here, we utilized trace fear conditioning to study the formation of trace fear memory in mice. We identified the entorhinal cortex (EC) as a critical component of sensory signaling to the amygdala. We adopted both loss-of-function and gain-of-function experiments to demonstrate that release of the cholecystokinin (CCK) from the EC is required for trace fear memory formation. We discovered that CCK-positive neurons project from the EC to the lateral nuclei of the amygdala (LA), and inhibition of CCK-dependent signaling in the EC prevented long-term potentiation of the auditory response in the LA and formation of trace fear memory. In summary, high-frequency activation of EC neurons triggers the release of CCK in their projection terminals in the LA, potentiating auditory response in LA neurons. The neural plasticity in the LA leads to trace fear memory formation.

**\*For correspondence:**
jufanghe@cityu.edu.hk

**Present address:** [†]Department of Neurosurgery, Stanford University School of Medicine, Stanford, United States; [‡]F.M. Kirby Neurobiology Center, Boston Children's Hospital, Boston, United States

**Competing interest:** The authors declare that no competing interests exist.

## Editor's evaluation

While the amygdala is important for associating innocuous sensory stimuli with aversive outcomes during associative fear learning, the medial temporal lobe memory system, including the entorhinal cortex, participates in bridging temporal gaps (trace periods) between the sensory stimuli and aversive outcomes. However, the circuit connections between these structures that allow for trace fear learning have not been clarified. Here, Feng et al. reveal that a specific population of cholecystokinin cells in the entorhinal cortex that project to the lateral nucleus of the amygdala are important for trace fear memory formation.

## Introduction

Learning to associate environmental cues with subsequent adverse events is an important survival skill. Fear conditioning is widely used to study this association and is performed by pairing a neutral stimulus (conditioned stimulus, CS), such as a tone, with a punishing stimulus (unconditioned stimulus, US), such as a shock (*Estes and Skinner, 1941*). The CS-US pair elicits fear behaviors, including freezing and fleeing, which are often species-specific. Canonical delay fear conditioning is performed by terminating the CS and US at the same time. However, CS and US do not necessarily occur simultaneously in nature, and the brain has evolved mechanisms to associate temporally distinct events.

Trace fear conditioning is used to study these mechanisms by inserting a trace interval between the end of the CS and the beginning of the US. The temporal separation between the CS and the US substantially increases the difficulty of learning as well as the recruitment of brain structures (*Crestani et al., 2002*; *Runyan et al., 2004*). Although trace fear conditioning provides essential insight into the neurobiology of learning and memory, many unanswered questions remain. For instance, the detailed neural circuitry underlying the formation of this trace fear memory and the potential modulatory chemicals involved in this process need to be further characterized.

Synaptic plasticity is the basis of learning and memory and refers to the ability of neural connections to become stronger or weaker. Long-term potentiation (LTP) is one of the most widely studied forms of synaptic plasticity. The lateral nucleus of the amygdala (LA) receives multi-modal sensory inputs from the cortex and thalamus. It relays them into the central nucleus of the amygdala, which then innervates the downstream effector structures (*Phelps and LeDoux, 2005*). LTP is developed in the auditory input pathway that signals to the LA. Auditory-responsive units in the LA fire faster after auditory-cued fear conditioning (*Quirk et al., 1995*). Optogenetic manipulation of the auditory input terminals in the LA leads to the suppression or recovery of LTP in the LA and can correspondingly suppress or recover conditioned fear responses (*Nabavi et al., 2014*). Researchers recently discovered that synaptic plasticity can occur upstream of the LA (*Barsy et al., 2020*), providing new insights into this fundamental topic. Nevertheless, synaptic plasticity in the LA is impressively correlated with the formation of fear memory.

Besides the amygdala, the hippocampus (*Bangasser et al., 2006*; *Gilmartin et al., 2012*), anterior cingulate cortex (*Han et al., 2003*), medial prefrontal cortex (mPFC) (*Runyan et al., 2004*; *Gilmartin and Helmstetter, 2010*), and entorhinal cortex (EC) (*Ryou et al., 2001*) are also involved in trace fear conditioning. The EC is integrated with the spatial and navigation systems of the animal (*Fyhn et al., 2004*; *Hafting et al., 2005*) and is essential for context-related fear associative memory (*Maren and Fanselow, 1997*). Moreover, the EC functions as a working memory buffer in the brain to hold information for temporal associations (*Fransén, 2005*; *Schon et al., 2016*). Here, a scenario of the dependence on the EC to associate the temporally separated CS and US is manifested.

Cholecystokinin (CCK) is the most abundant neuropeptide in the central nervous system (CNS) (*Rehfeld, 1978*). CCK has two recognized receptors in the CNS: CCK A receptor (CCKAR) and CCK B receptor (CCKBR). Previous studies in our laboratory unveiled that CCK and CCKBR enabled neuroplasticity as well as associative memory between two sound stimuli and between visual and auditory stimuli (AS) in the auditory cortex (AC) (*Li et al., 2014*; *Chen et al., 2019*; *Zhang et al., 2020*). CCK and its receptors are intrinsically involved in fear-related mental disorders including anxiety (*Chen et al., 2006*), depression (*Shen et al., 2019*), and post-traumatic stress disorder (PTSD) (*Joseph et al., 2013*). Moreover, the CCKBR agonist CCK-tetrapeptide (CCK-4) induces acute panic attacks in healthy human subjects and patients with a panic disorder (*Bradwejn, 1993*). Despite the clear connection between CCK and fear-related disorders, it remains elusive the involvement of CCK in fear conditioning and the formation of cue-specific fear memory, which is possibly the neural foundation of these disorders.

In the present study, we investigated the involvement of CCK-expressing neurons in the EC in trace fear memory formation. We then examined how CCK enabled neuroplasticity in the auditory pathway to the LA by conducting the in vivo recording in the LA. Finally, we studied the contribution of the EC to LA pathway on the formation of trace fear memory in the physiological and behavioral context.

## Results

### Loss of CCK results in deficient trace fear memory formation in *Cck*<sup>-/-</sup> mice

The first question we asked here was whether CCK is involved in trace fear memory formation. We studied transgenic *Cck*<sup>-/-</sup> mice (*Cck*-CreER, strain #012710, Jackson Laboratory), which lack CCK expression (*Chen et al., 2019*). We subjected *Cck*<sup>-/-</sup> and wildtype (WT) control (C57BL/6) mice to trace fear conditioning using two training protocols: long-trace interval and short-trace interval training.

We performed the trace fear conditioning experiment by collecting baseline readouts on pre-conditioning day, training with the appropriate CS-US pairings on conditioning days, and testing the conditioned fear responses on post-conditioning/testing day. In the long trace protocol, mice

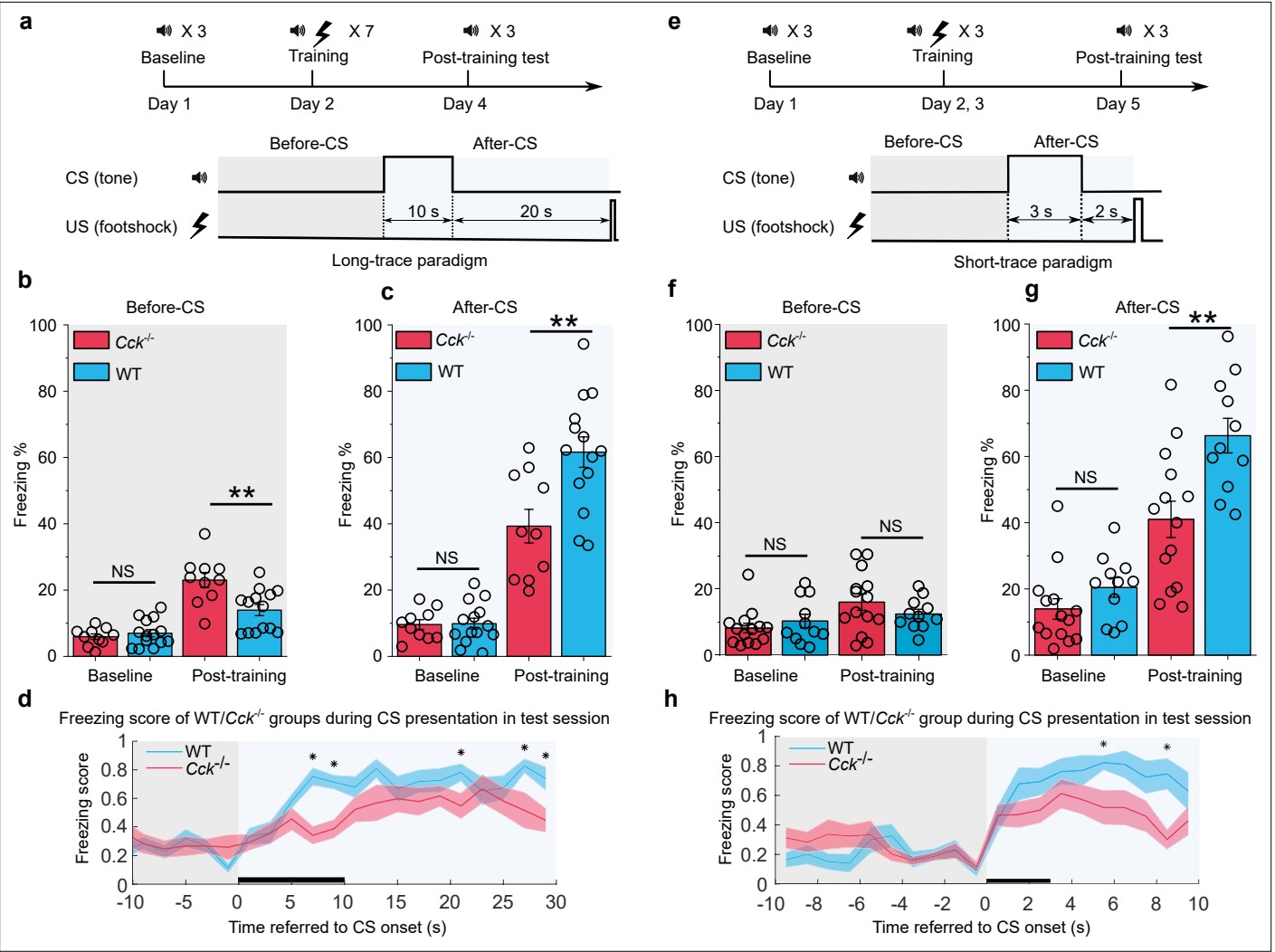

**Figure 1.** Trace fear memory formation deficit in *Cck*⁻/⁻ mice. (**a**) Schematic diagram of the fear conditioning paradigm with a long trace interval of 20 s. Gray and light blue shadowed areas indicate the time frames before and after the onset of the CS (before-CS, after-CS). CS, conditioned stimulus; US, unconditioned stimulus. (**b–c**) Freezing percentages before (**b**) and after (**c**) the CS. Freezing percentages were recorded at baseline on the pre-conditioning day and post-training on the post-conditioning day. WT, wildtype, N = 14; *Cck*⁻/⁻, CCK-knockout, N = 10. *p < 0.05; **p < 0.01; ***p < 0.001; NS, not significant. Statistical significance was determined by two-way RM ANOVA with Bonferroni post hoc pairwise comparison. RM ANOVA, repeated-measures analysis of variance. (**d**) Freezing score plot of the two groups of mice during the testing session. The freezing score was binned in a 2 s interval. Solid lines indicate the mean value, and shadowed areas indicate the SEM. The black bar indicates the presence of the CS from 0 to 10 s. Two-way RM ANOVA with a Greenhouse-Geisser correction, interaction significant, $F_{(8.214, 180.716)} = 2.149$, $p < 0.05$; post hoc Bonferroni multiple pairwise comparisons between two groups in each bin, *p < 0.05. SEM, standard error of the mean. (**e**) Schematic diagram of the fear conditioning paradigm with a short-trace interval of 2 s. (**f–g**) Freezing percentages before (**f**) and after (**g**) the CS. WT, N = 11; *Cck*⁻/⁻, N = 14. (**h**) Freezing score plot of the two groups of mice during the testing session. Freezing score was binned in a 1 s interval. The black bar indicates the presence of the CS from 0 to 3 s. Two-way RM ANOVA with a Greenhouse-Geisser correction, interaction significant, $F_{(8.093, 186.145)} = 2.499$, $p < 0.05$; post hoc Bonferroni multiple comparisons in each bin, *p < 0.05.

The online version of this article includes the following source data and figure supplement(s) for figure 1:

**Source data 1.** Summary of freezing percentage in long and short trace fear conditioning.

**Figure supplement 1.** Genetic and behavioral examination of *Cck*⁻/⁻ mice.

sequentially received a 10 s pure tone (as the CS), a 20 s gap (trace interval), and a 0.5 s foot shock (as the US) (*Figure 1a*). We calculated the percentage of time frames where mice displayed a freezing response as the measure of fear memory. Freezing percentages were compared before (baseline) and after (post-training) trace fear conditioning as well as before (*Figure 1b*) and after (*Figure 1c*) presentation of the CS. The after-CS freezing percentage was calculated within the time window that

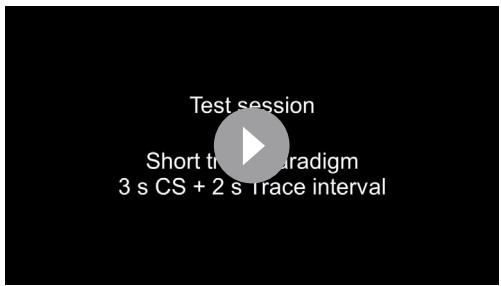

**Video 1.** Freezing response of wildtype (WT) mice to the conditioned stimulus (CS) in the test session after long-trace fear conditioning paradigm, related to Figure 1b–c. WT mice showed significant freezing response to the CS after training.
https://elifesciences.org/articles/69333/figures#video1

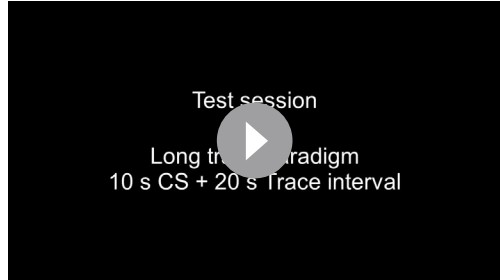

**Video 3.** Freezing response of wildtype (WT) mice to the conditioned stimulus (CS) in the test session after short-trace fear conditioning paradigm, related to Figure 1f–g. WT mice showed significant freezing response to the CS after training.
https://elifesciences.org/articles/69333/figures#video3

includes the duration of CS (10 s) and the trace interval (20 s). For before-CS freezing percentage, we selected the time window with same length (30 s) just before the presentation of CS. At baseline, $Cck^{-/-}$ (N = 10/2 cages) and WT (N = 14/3 cages) mice showed similarly low freezing percentages both before (*Figure 1b*) and after (*Figure 1c*) the CS (*Figure 1b*, two-way repeated-measures analysis of variance [RM ANOVA], significant interaction, F[1,22] = 10.85, p = 0.003 < 0.01; pairwise comparison, WT vs. $Cck^{-/-}$ before CS, 7.0% ± 1.0% vs. 5.9% ± 1.1%; 95% confidence interval [CI], [5.0%, 9.0%] vs. [3.6%, 8.3%]; Bonferroni test, p = 0.482 > 0.05; *Figure 1c*, two-way RM ANOVA, significant interaction, F[1,22] = 8.94, p = 0.007 < 0.01; pairwise comparison, WT vs. $Cck^{-/-}$ after CS, 9.9% ± 1.5% vs. 9.6% ± 1.8%; 95% CI, [6.8–13.0%] vs. [5.9–13.3%]; Bonferroni test, p = 0.911 > 0.05). After conditioning, $Cck^{-/-}$ mice showed significantly lower freezing percentages (39.3% ± 5.3%, 95% CI, [28.3%, 50.2%]) than WT mice after receiving the CS (61.6% ± 4.5%, 95% CI, [52.4%, 70.9%]; pairwise comparison, p = 0.004 < 0.01), indicating poor performance in associating the CS with the US (*Figure 1c*, *Videos 1 and 2*). This effect was not due to elevated basal freezing levels caused by training in WT animals. Instead, we found that $Cck^{-/-}$ mice (23.0% ± 2.1%, 95% CI, [18.6%, 27.4%]) had higher freezing percentages than WT mice (14.0% ± 1.8%, 95% CI, [10.3%, 17.7%]) in the absence of the CS (*Figure 1b*, pairwise comparison, p = 0.003 < 0.01). Together, these results suggest that trace fear conditioning results in elevated conditioned freezing percentages in WT mice, which are primarily elicited by the CS, and that loss of CCK impairs the freezing response to the CS. Furthermore, we defined an empirical threshold of moving velocity and converted the moving velocity to a binary freezing score plot, in which value 1 represents active state, and value 0 represents freezing state (see Materials and methods). Using this method, we were able to assess the freezing response of the animal as it occurred during the CS presentation. Again, we found that WT mice obtained higher average

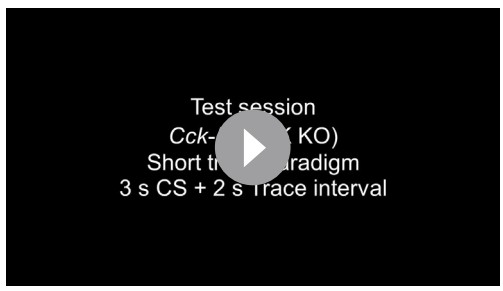

**Video 2.** Freezing response of $Cck^{-/-}$ mice to the conditioned stimulus (CS) in the test session after long-trace fear conditioning paradigm, related to Figure 1b–c. $Cck^{-/-}$ mice showed impaired freezing response to the CS after training.
https://elifesciences.org/articles/69333/figures#video2

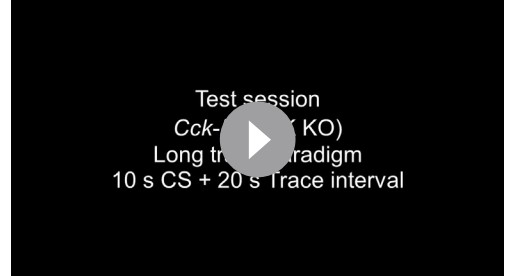

**Video 4.** Freezing response of $Cck^{-/-}$ mice to the conditioned stimulus (CS) in the test session after short-trace fear conditioning paradigm, related to Figure 1f–g. $Cck^{-/-}$ mice showed impaired freezing response to the CS after training.
https://elifesciences.org/articles/69333/figures#video4

freezing scores than *Cck*-/- mice during and after the presentation of the CS (*Figure 1d*, two-way RM ANOVA with a Greenhouse-Geisser correction, interaction significant, F(8.214, 180.716) = 2.149, p = 0.032 < 0.05; post hoc Bonferroni multiple pairwise comparisons between two groups in each bin, *p = 0.00015, 0.00036, 0.031, 0.015, 0.022 < 0.05 at time point 6–8, 8–10, 20–22, 26–28, and 28–30 s referred to the onset of CS, respectively).

In addition to the long-trace interval, we also investigated freezing responses of mice during a short-trace fear conditioning paradigm. Mice were presented a 3 s CS followed by a 2 s trace interval and a 0.5 s electrical foot shock (*Figure 1e*). Same as above, freezing percentage in the after-CS period was calculated from the time window that includes duration of the CS (3 s) and the trace interval (2 s), and before-CS freezing percentage was from a 5-s-long time window right before the presentation of the CS. Before training, WT (N = 11/3 cages) and *Cck*-/- (N = 14/4 cages) mice showed similarly low freezing percentages both before (*Figure 1f*) and after (*Figure 1g*) presentation of the CS (*Figure 1g*, two-way RM ANOVA, significant interaction, F[1,23] = 5.18, p = 0.032 < 0.05; pairwise comparison, WT vs. *Cck*-/- in the baseline session, 20.4% ± 3.3% vs. 13.9% ± 2.9%; 95% CI, [13.7%, 27.2%] vs. [8.0%, 19.9%]; p = 0.150 > 0.05; *Figure 1f*, two-way RM ANOVA, interaction not significant, F[1,23] = 1.99, p = 0.17 > 0.05; pairwise comparison, WT vs. *Cck*-/- in the baseline session, 10.3% ± 1.8% vs. 8.2% ± 1.6%; 95% CI, [6.5%, 14.1%] vs. [4.8%, 11.6%]; p = 0.402 > 0.05). Consistent with results from the long-trace paradigm, *Cck*-/- mice showed an impaired freezing response (41.0% ± 5.1%) to the CS after training compared to WT mice (66.3% ± 5.2%; 95% CI, [54.3%, 78.3%]; pairwise comparison, p = 0.003 < 0.01, *Figure 1g*, *Videos 3–4*). Additionally, we observed no significant difference between fear conditioned WT and *Cck*-/- mice prior to the presentation of the CS (*Figure 1f*, pairwise comparison, WT vs. *Cck*-/- in the post-training session, 12.4% ± 2.3% vs. 16.0% ± 2.0%; 95% CI, [7.7%, 17.2%] vs. [11.8%, 20.2%]; p = 0.253 > 0.05). Finally, we found significant differences in freezing scores between WT and *Cck*-/- mice when presented the CS (*Figure 1h*, two-way RM ANOVA with a Greenhouse-Geisser correction, interaction significant, F(8.093, 186.145) = 2.499, p = 0.013 < 0.05; post hoc Bonferroni multiple comparisons in each bin, *p = 0.034, 0.001 < 0.05 at time point 5–6 and 8–9 s referred to the onset of the CS, respectively).

We conducted the innate hearing and fear expression examinations to rule out a potential inherent deficit derived from genome editing in *Cck*-/- transgenic mice. To evaluate hearing, we recorded the open-field auditory brainstem response (ABR) in anesthetized animals. We observed five peaks in both WT and *Cck*-/- mice at sound intensities above 50 dB of sound pressure level (dB SPL) (*Figure 1—figure supplement 1b*), and we did not observe any remarkable differences between the waveforms. Compared to WT mice, *Cck*-/- mice had better hearing (40.0 ± 1.2 dB in *Cck*-/- mice, N = 15/3 cages, vs. 47.3 ± 2.1 dB in WT mice, N = 11/3 cages, two-sample t-test, t(24) = 3.238, p = 0.003 < 0.01, *Figure 1—figure supplement 1c*). Thus, auditory perception does not account for the deficient trace fear memory formation of *Cck*-/- mice.

Fear expression is the behavioral output of fear conditioning. We wondered if *Cck*-/- mice suffered from a deficit in fear expression, which is observed in Klüver-Bucy syndrome and other diseases (*Lilly et al., 1983*). To test whether the *Cck*-/- mice have a deficit in fear expression, we presented a loud (90 dB SPL) white noise and quantified sound-driven innate freezing. We found no statistical difference between WT (46.1% ± 5.5%, N = 11/3 cages) and *Cck*-/- mice (46.5% ± 6.6%, N = 14/3 cages, two-sample t-test, t(23) = 0.046, p = 0.964 > 0.05, *Figure 1—figure supplement 1d*), indicating that *Cck*-/- mice can express passive defensive behaviors such as freezing. Thus, the deficiency in trace fear memory formation of *Cck*-/- is not due to a deficit in fear expression. It may be due to a deficit in establishing an association between the CS and the US.

In summary, *Cck*-/- mice display deficient trace fear memory formations in both short- and long-trace models that are not caused by inherent hearing or fear expression abnormalities.

## Deficient neural plasticity in the LA of *Cck*-/- mice

As neural plasticity in the LA is widely regarded as the basis of fear memory formation (*Rogan et al., 1997*; *LeDoux, 2000*; *Nabavi et al., 2014*; *Kim and Cho, 2017*), we examined LTP in the LA of WT and *Cck*-/- mice by in vivo recording (*Figure 2a*). First, we successfully recorded the auditory evoked potential (AEP) in the LA of anesthetized WT and *Cck*-/- mice (*Figure 2b–e*). Then, we used theta-burst electrical stimulation to induce LTP of AEP (AEP-LTP) (*Figure 2f*). Interestingly, AEP-LTP was effectively induced in WT mice (N = 15/6 cages) but was not in *Cck*-/- mice (N = 12/4 cages). WT

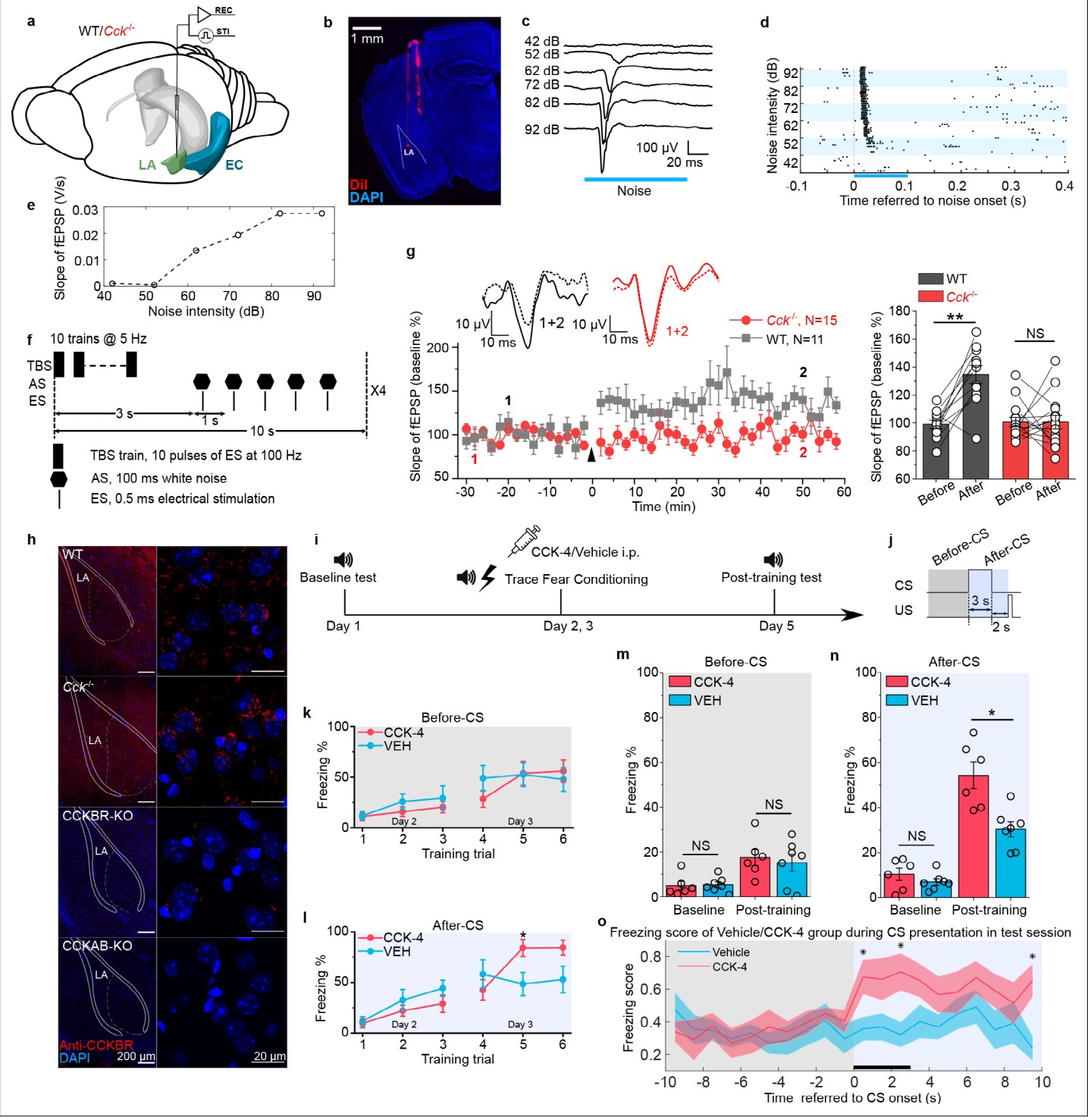

**Figure 2.** Neural plasticity deficit in the LA of *Cck⁻/⁻* mice and the rescuing effect of exogenous cholecystokinin (CCK). (**a**) Schematic diagram of in vivo recording in the LA. EC, entorhinal cortex; LA, lateral amygdala. STI, stimulation. REC, recording. (**b**) Post hoc verification of electrode tracks and recording area. (**c**) Representative AEP traces in response to different levels of noise stimulus. AEP, auditory evoked potential. (**d**) Representative traces of multiunit spikes to different levels of noise stimulus. (**e**) Representative input/output (I/O) curve of the slope of AEP vs. noise intensity. fEPSP, field excitatory postsynaptic potential. (**f**) Schematic diagram of the pairing protocol to induce LTP of AEP via theta-burst stimulation (TBS). LTP, long-term potentiation; ES, electrical stimulation; AS, auditory stimulation. (**g**) Time course plot of the normalized AEP slope during LTP. The wildtype (WT) group is indicated in black, and the *Cck⁻/⁻* in red. Representative traces of the AEP before (dotted line) and after (solid line) TBS are shown in inset panels for both groups. The average normalized slopes 10 min before pairing (−10–0 min, before) and 10 min after pairing (50–60 min, after) in the two groups of

*Figure 2 continued on next page*

*Figure 2 continued*

mice are shown on the right. **p < 0.01; two-way RM ANOVA with post hoc Bonferroni pairwise comparison; RM ANOVA, repeated-measures analysis of variance; NS, not significant. (**h**) Immunofluorescent staining of CCK B receptor (CCKBR) in brain slices from WT, *Cck*[-/-], CCKBR-KO, and CCKAB-KO mice. Magnified images are shown on the right. CCKBR-KO, CCK B receptor knock-out mouse; CCKAB-KO, CCK A receptor and B receptor double knock-out mouse. (**i**) Experimental timeline for (j–o). (**j**) Schematic diagram of the CS-US presentation. Gray and light blue shadowed areas indicate the time frames before and after CS presentation (before-CS, after-CS). (**k–l**) Freezing percentages before (**k**) and after (**l**) the CS during fear conditioning training on training day. Animals underwent six trials during a 2-day training (days 2 and 3). Two-way RM ANOVA with Bonferroni pairwise comparison, *p < 0.05. (**m–n**) Freezing percentages before (**m**) and after (**n**) the CS on the pre-training day (baseline) and the post-training day. CCK-4, N = 6; VEH, N = 7; *p < 0.05; NS, not significant; two-way RM ANOVA with Bonferroni post hoc pairwise test; RM ANOVA, repeated-measures analysis of variance. (**o**) Freezing score plot of the two groups of mice during the testing session on day 5. Solid lines indicate the mean value, and shadowed areas indicate the SEM. The black bar indicates the presence of the CS from 0 to 3 s. Two-way RM ANOVA with a Huynh-Feldt correction, interaction significant, F(17.22, 189.417) = 1.932, p = 0.017 < 0.05; post hoc Bonferroni multiple comparisons, *p < 0.05.

The online version of this article includes the following source data and figure supplement(s) for figure 2:

**Source data 1.** AEP-LTP induction in WT and CCK-KO mice.

**Figure supplement 1.** Exogenous cholecystokinin-tetrapeptide (CCK-4) activates CCK B receptor (CCKBR) in the lateral nuclei of the amygdala (LA).

mice demonstrated remarkable potentiation (*Figure 2g*, two-way RM ANOVA, significant interaction, F[1,25] = 6.775, p = 0.015 < 0.05; pairwise comparison, after vs. before induction, 142.7% ± 12.6% vs. 99.1% ± 3.4%, p = 0.003 < 0.01), whereas *Cck*[-/-] mice showed no potentiation (pairwise comparison, after vs. before induction, 98.0% ± 11.3% vs. 100.6% ± 3.0%, p = 0.824 > 0.05). These results suggest that *Cck*[-/-] mice have a deficit in neural plasticity in the LA that may contribute to their reduced response to trace fear conditioning.

## Stimulation of CCKBR rescues the formation of trace fear memory in *Cck*[-/-] mice

Although the translation and release of CCK are disrupted in *Cck*[-/-] mice, we found that the predominant CCK receptor, CCKBR, was expressed normally in both WT and *Cck*[-/-] mice (*Figure 2h*). Therefore, we hypothesized that exogenous stimulation of CCKBR might rescue trace fear memory deficits in *Cck*[-/-] mice. CCKBR can be stimulated by several agonists, including CCK octapeptide sulfated (CCK-8s) and CCK tetrapeptide (CCK-4). As CCK-8s is a potent agonist of both CCKAR and CCKBR, we selected CCK-4, which is a preferred CCKBR agonist (*Berna et al., 2007*). To monitor CCK signaling in vivo, we expressed a G protein-coupled receptor (GPCR) activation-based CCK sensor (GRAB_CCK, AAV-hSyn-CCK2.0) in the LA of WT mice (*Jing et al., 2019*). Using this model, binding of the GPCR CCKBR with endogenous or exogenous CCK results in increased fluorescence intensity, which we measured by fiber photometry in the LA (*Figure 2—figure supplement 1a*). We first confirmed that intraperitoneal (i.p.) administration of CCK-4 penetrated the blood-brain barrier (BBB) and activated the CCK2.0 sensor. Moreover, we demonstrated that the administration of CCK-4 evoked an apparent and long-term increase in the fluorescent signal (*Figure 2—figure supplement 1b-c*). Together, these data verify that CCK-4 passes through the BBB and binds with CCKBR in the LA.

After validating our model, we conducted short-trace fear conditioning in *Cck*[-/-] mice on 2 consecutive days just after intraperitoneal administration of CCK-4 or the corresponding vehicle (VEH) (*Figure 2i–j*). We collected data during the 2 conditioning days to monitor the learning curve of mice as conditioning progressed. The learning curves were plotted as the freezing percentages of CCK-4- or VEH-treated *Cck*[-/-] mice during the six training trials (*Figure 2k–l*). We did not observe any statistical differences between the two groups during the first three trials on the first conditioning day and even in the fourth trial on the second conditioning day. We found that CCK-4-treated mice had significantly higher freezing

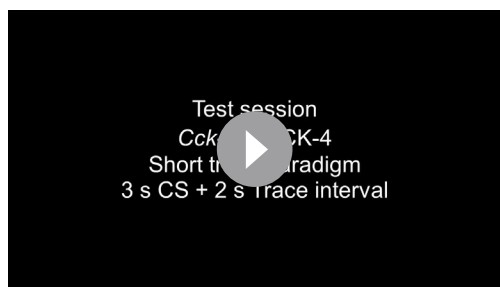

**Video 5.** Freezing response to the conditioned stimulus (CS) of *Cck*[-/-] mice treated with cholecystokinin tetrapeptide (CCK-4) in the test session after short-trace fear conditioning paradigm, related to Figure 2m–n. CCK-4-treated mice showed significant freezing response to the CS after training.
https://elifesciences.org/articles/69333/figures#video5

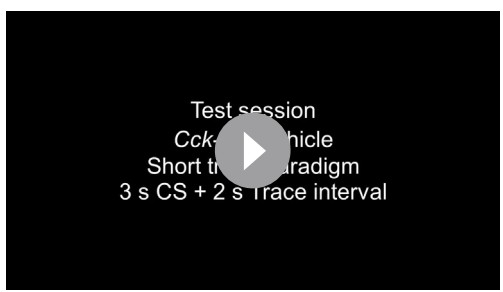

**Video 6.** Freezing response to the conditioned stimulus (CS) of *Cck^-/-* mice treated with vehicle in the test session after short-trace fear conditioning paradigm, related to Figure 2m–n. Vehicle-treated mice showed impaired freezing response to the CS after training.
https://elifesciences.org/articles/69333/figures#video6

levels than VEH-treated mice during the fifth training trials conducted on the second conditioning day (*Figure 2l*, two-way RM ANOVA, interaction significant, F[5, 65] = 3.45, p = 0.008 < 0.01; Bonferroni pairwise comparison, 84.2% ± 8.4% in the CCK-4 group [N = 7/2 cages] vs. 48.4% ± 11.5% in the VEH group [N = 8/2 cages] in the fifth trial, p = 0.029 < 0.05; 84.4% ± 7.3% in the CCK-4 group vs. 52.9% ± 13.0% in the VEH group in the sixth trial, p = 0.064). In support of this evidence, we did not find a statistical difference between the two groups prior to CS presentation during the fifth or sixth trials (*Figure 2k*, two-way RM ANOVA, F[5, 65] = 0.696, p = 0.628 > 0.05; Bonferroni pairwise comparison, 53.8% ± 11.5% in the CCK-4 group vs. 52.5% ± 11.8% in the VEH group in the fifth trial, p = 0.938 > 0.05; 56.0% ± 10.8% in the CCK-4 group vs. 47.8% ± 11.8% in the VEH group in the sixth trial, p = 0.622 > 0.05). Together, these data suggest that mice in the CCK-4- and VEH-treated groups showed similar baseline freezing levels and that CCK-4 treatment improved trace fear conditioning learning responses in *Cck^-/-* mice.

We examined the conditioned fear response in CCK-4- and VEH-treated *Cck^-/-* mice 2 days after training compared to fear responses at baseline before training (*Figure 2m–n*). We found that CCK-4-treated mice showed remarkably higher freezing levels than VEH-treated mice post-training, whereas no significant difference was detected at baseline (*Figure 2n*, two-way RM ANOVA, significant interaction, F[1,11] = 6.40, p = 0.028 < 0.05; pairwise comparison, CCK-4 vs. VEH at baseline, 10.4% ± 2.2% vs. 7.0% ± 2.0%; 95% CI, [5.6%, 15.2%] vs. [2.5%, 11.5%]; p = 0.278 > 0.05; CCK-4 vs. VEH post-training, 54.3% ± 4.8% vs. 30.4% ± 4.4%; 95% CI, [43.8%, 64.8%] vs. [20.6%, 40.1%]; p = 0.004 < 0.05; *Videos 5–6*). There was no statistical difference between the two groups before the presentation of the CS (*Figure 2m*, two-way RM ANOVA, interaction not significant, F[1, 11] = 0.174, p = 0.684 > 0.05; the main effect of drug application [CCK-4 vs. VEH] on freezing percentage was not significant, F[1,11] = 0.15, p = 0.706 > 0.05). Additionally, CCK-4-treated mice had significantly higher freezing scores than VEH-treated mice (*Figure 2o*, two-way RM ANOVA with a Huynh-Feldt correction, interaction significant, F(17.22, 189.417) = 1.932, p = 0.017 < 0.05; post hoc Bonferroni multiple comparisons, *p = 0.025, 0.014, 0.005 < 0.05 at time point 0–1, 2–3, 9–10 s referred to the onset of the CS, respectively). These results indicate that CCK-4 treatment effectively improved learning response to trace fear conditioning in *Cck^-/-* mice. Moreover, this rescue was not an artifact caused by reduced locomotion after drug application and fear conditioning training, as there was no difference between the two groups in the freezing percentage prior to presentation of the CS (*Figure 2m*). Therefore, the exogenous application of a CCKBR agonist activated endogenous CCKBR and improved the fear memory formation of *Cck^-/-* mice after trace fear conditioning.

## CCK neurons in the EC are critical for the formation of the trace fear memory

We next examined the source of endogenous CCK that signals to the LA. We injected a potent retrograde neuronal tracer cholera toxin subunit B (CTB) conjugated to a fluorescent tag Alexa-647 (CTB-647) into the LA and dissected the upstream anatomical brain regions that project to the LA (*Figure 3a*). In addition to regions that are canonically involved in fear circuitry, including the AC and the medial geniculate body (MGB), we found that EC was also densely labeled with retrograde CTB-647, suggesting that the EC is connected with the LA (*Figure 3b–e*). We next injected a Cre-dependent retrograde AAV (retroAAV-hSyn-FLEX-jGcamp7s) into the LA of CCK-ires-Cre (CCK-Cre) mice to label CCK-positive neurons that project into the LA, further to confirm the above observation (*Figure 3f–g*). In the CCK-ires-Cre mouse line, Cre expression was restricted to the CCK-expressing neurons, where the Cre-mediated recombination took place and the Cre-dependent green fluorescent

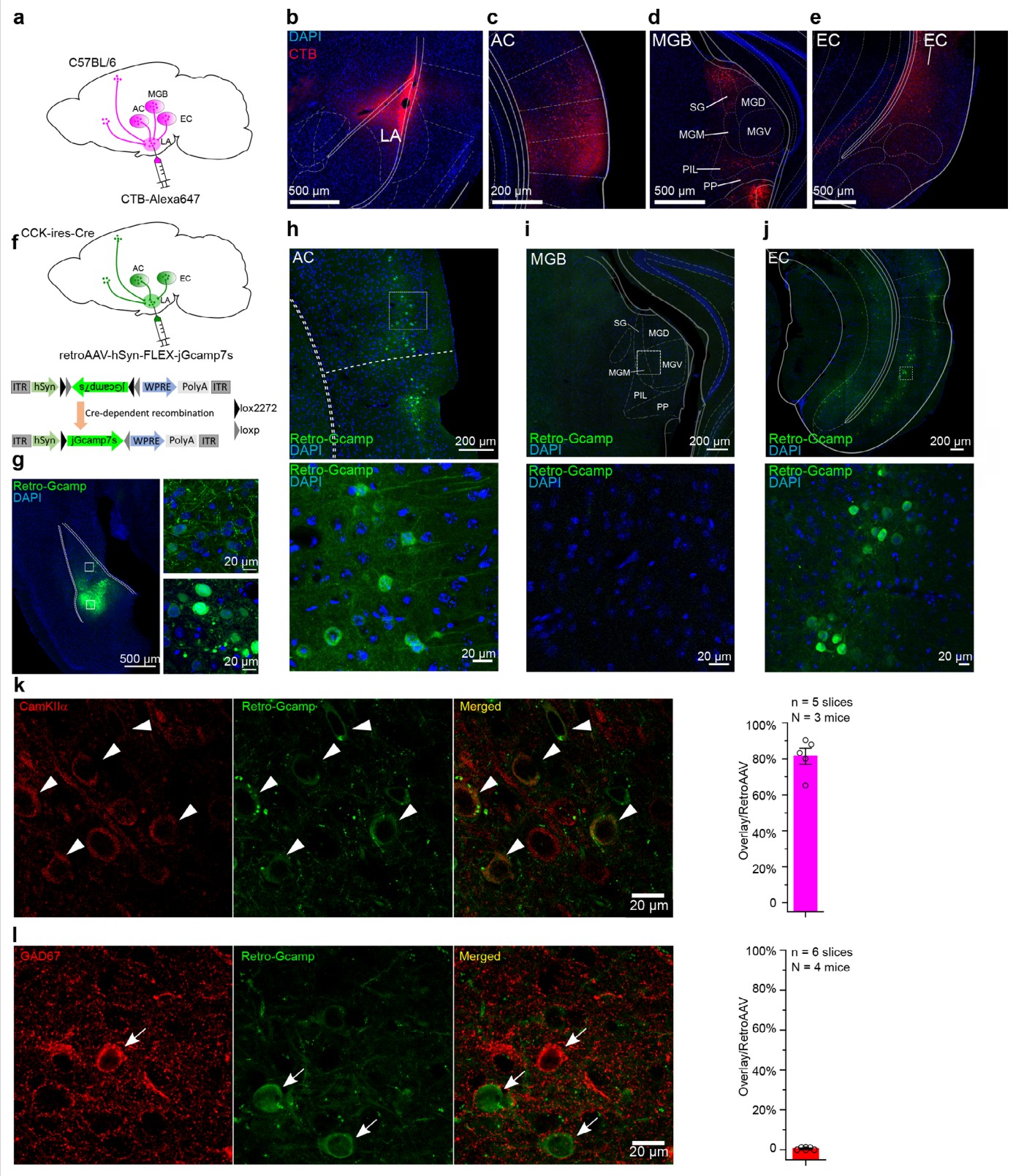

**Figure 3.** Dissection of inputs of the lateral nuclei of the amygdala (LA) with retrograde tracer and virus. (**a**) Schematic diagram of retrograde tracing with Alexa647-conjugated cholera toxin subunit B (CTB). (**b–e**) Representative fluorescent images of the injection site of the CTB tracer (**b**), the canonical upstream regions, including the auditory cortex (**c**) and the auditory thalamus (**d**), and the non-canonical entorhinal cortex (**e**). AC, auditory cortex; MGB, medial geniculate body; SG, suprageniculate thalamic nucleus; MGM, medial MGB; PIL, posterior intralaminar thalamic nucleus; PP, peripeduncular

*Figure 3 continued on next page*

Figure 3 continued

nucleus; EC, entorhinal cortex. (**f**) Schematic diagram of cell type-specific retrograde tracing with Cre-dependent retrograde AAV (retroAAV-hSyn-FLEX-jGcamp7s). (**g**) Verification of the injection site in the LA. Magnified images are shown in insets on the right. Retro-Gcamp, retrograde jGcamp7s signal. (**h–j**) Retrograde signals in the AC (**h**), MGB (**i**), and EC (**j**). Magnified images are shown in the bottom insets. (**k–l**) Co-immunofluorescent staining of retrograde tracing of the LA with either the excitatory neuronal marker CamKIIα (**k**) or the inhibitory neuronal marker GAD67 (**l**). Bar charts showing the proportion of CamKIIα or GAD67-positive neurons in retroAAV-labeled ones are placed in the right panel accordingly.

The online version of this article includes the following source data and figure supplement(s) for figure 3:

**Source data 1.** Summary of colocalization between Retro-Gcamp and CamKIIα or GAD67.

**Figure supplement 1.** Retrograde labeling of inputs of the lateral nuclei of the amygdala (LA).

protein jGcamp7s was expressed (*Figure 3f*). Fluorescent signal was detected in the AC and the EC, but not in the MGB (*Figure 3h–j*), which suggests that CCK may originate from these two brain regions during trace fear memory formation. Immunofluorescent staining revealed that most CCK-positive neurons in the EC that project to the LA are glutamatergic (*Figure 3k–l*), which is consistent with our previous findings in CCK-positive neurons in the EC (*Chen et al., 2019*).

Considering the potential tropism of retroAAV that may cause the absence of AAV expression in the MGB, we injected a Cre-expressing retroAAV (retroAAV-hSyn-Cre) into the LA of the Cre-dependent tdTomato reporter Ai14 mice (N = 3/1 cage). Besides the AC and EC, we also found the tdTomato-positive neurons in the MGB suggesting retroAAV does not have the tropism to avoid expression in the MGB (*Figure 3—figure supplement 1*). However, based on our ongoing studies, we cannot exclude the possible scenario that MGB may originate some CCK-positive projection to LA during some stages of development.

Interestingly, the EC is involved in the formation of trace fear memory but is not a component of canonical delay fear memory (*Esclassan et al., 2009*). This selectivity suggests that the EC may be a component of the neural circuit underlying trace fear memory formation. To evaluate a requirement for the EC in trace fear memory, we utilized a designer receptors exclusively activated by designer drugs (DREADD) system to silence EC neurons (*Armbruster et al., 2007*). Specifically, the inhibitory receptor hM4Di was expressed in the EC of WT mice (*Figure 4a*) and was activated by administrating the designer drug clozapine (CLZ). Activation of hM4Di by CLZ induces membrane hyperpolarization, effectively silencing neurons. We verified EC neuron silencing by in vivo electrophysiological recording (*Figure 4b–d* and *Figure 4—figure supplement 1*). We found that a low dose of CLZ (0.5 mg/kg) effectively suppressed both instant and long-term neuronal firing. Of note, we used CLZ instead of the canonical DREADD ligand clozapine-N-oxide (CNO). A recent study identified CLZ as the active metabolite of CNO (*Gomez et al., 2017*), and CLZ more effectively penetrates the BBB and binds with DREADD receptors compared to CNO. As a result, a much lower dose of CLZ can elicit similar behavioral effects as higher doses of CNO (*Gomez et al., 2017*). Therefore, we used a low dose of CLZ (0.5 mg/kg) in our experiments.

Six weeks after injection of AAV9-hSyn-hM4Di-EGFP or AAV9-hSyn-EGFP, we administered CLZ by intraperitoneal injection and conducted short-trace fear conditioning 30 min later. We repeated the CLZ treatment and trace fear conditioning the following day and tested conditioned fear responses 2 days after that. As expected, mice expressing hM4Di (hM4Di, N = 7/2 cages) showed significantly lower freezing percentages in response to the CS than those expressing the control virus (EGFP, N = 7/2 cages) post-training (*Figure 4f*, two-way RM ANOVA, significant interaction, $F[1,12] = 6.58$, p = 0.025 < 0.05; EGFP vs. hM4Di post-training, 68.1% ± 8.1% vs. 38.9% ± 8.1%, p = 0.026 < 0.05; *Videos 7–8*). No significant differences were observed between the two groups at baseline (*Figure 4f*, pairwise comparison, EGFP vs. hM4Di at baseline, 12.0% ± 3.2% vs. 15.0% ± 3.2%, p = 0.530 > 0.05) or prior to the CS (*Figure 4e*, two-way RM ANOVA, interaction not significant, $F[1, 12] = 0.029$, p = 0.869 > 0.05; pairwise comparison, EGFP vs. hM4Di post-training, 16.0% ± 4.3% vs. 16.4% ± 4.3%, p = 0.952 > 0.05).

As we have shown that CCK-positive neural projections extend from the EC to the LA, we transfected CCK-expressing neurons in the EC with a Cre-dependent hM4Di in CCK-Cre mice (*Figure 4h–j*). These mice received an i.p. injection of CLZ (N = 10/3 cages) or VEH (N = 10/3 cages) prior to long-trace fear conditioning. After training, mice injected with CLZ showed significantly lower freezing percentages than those injected with the VEH, whereas no statistical differences were observed at baseline or prior to the CS (*Figure 4l*, two-way RM ANOVA, significant interaction, $F[1,18] = 5.904$, p

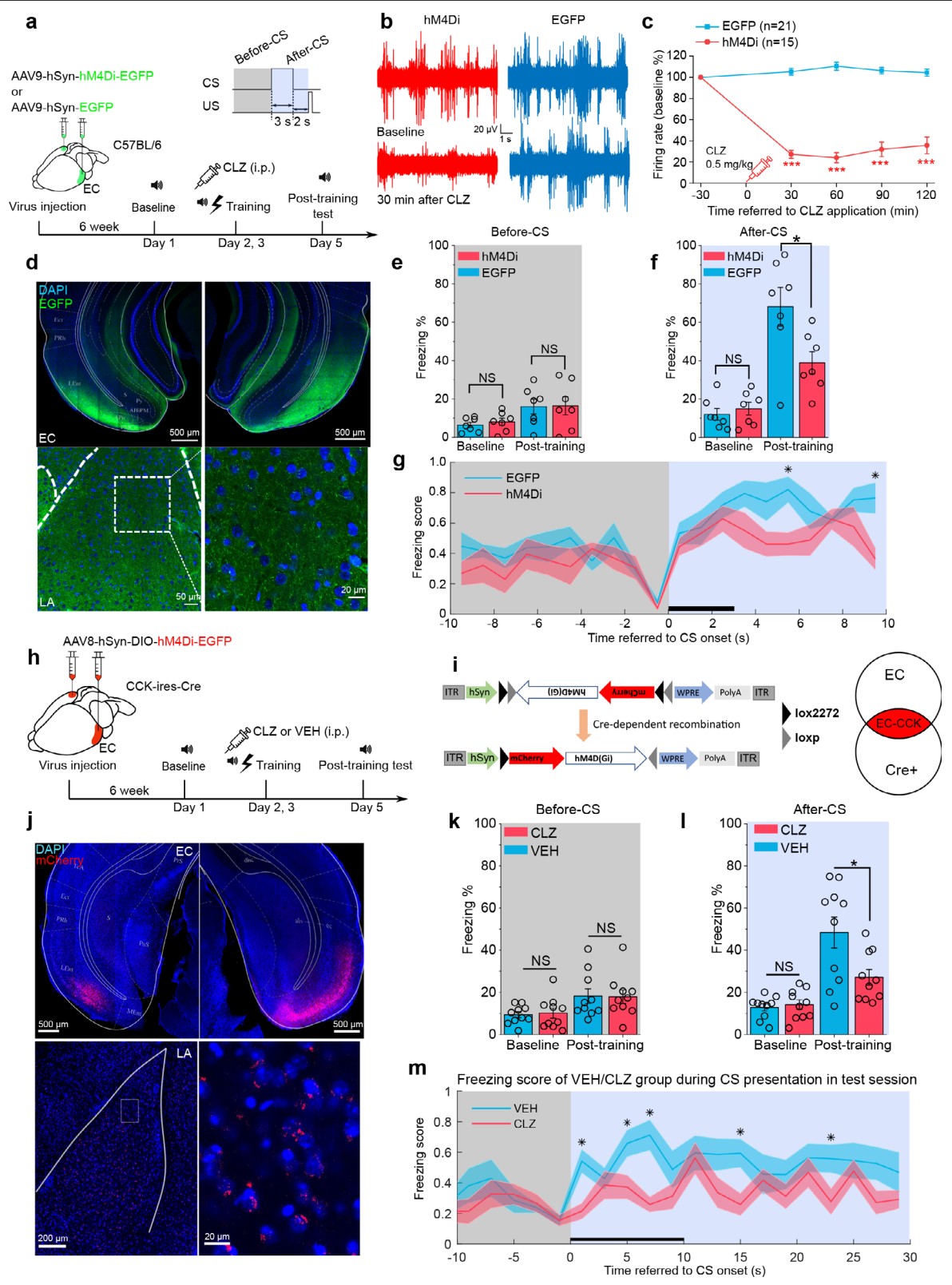

**Figure 4.** Formation of trace fear memory is suppressed by chemogenetic inhibition of the entorhinal cortex (EC) and cholecystokinin (CCK)-positive EC neurons. (**a**) Schematic diagram of trace fear conditioning and chemogenetic inhibition of the EC. EC, entorhinal cortex; hM4Di, inhibitory designer receptors exclusively activated by designer drugs (DREADD) receptor; CLZ, clozapine. (**b**) Representative traces of extracellular recording in the EC before and after systemic application of CLZ in hM4Di-expressing (red) and EGFP-expressing mice (blue). (**c**) Normalized firing rate of the EC neurons

*Figure 4 continued on next page*

*Figure 4 continued*

before and after systemic CLZ application. ***p < 0.001; two-sample t-test. (**d**) Verification of viral expression in the bilateral EC (top panel) and the EC-LA projection (bottom left panel). A magnified image of the EC-LA projection is shown in the bottom right inset. (**e–f**) Freezing percentages before (**e**) and after (**f**) the conditioned stimulus (CS) during the testing session in hM4Di-expressing (N = 7) or EGFP-expressing mice (N = 7). *p < 0.05; NS, not significant; two-way RM ANOVA with Bonferroni post hoc pairwise test; RM ANOVA, repeated-measures analysis of variance. (**g**) Freezing score plot of hM4Di-expressing and EGFP-expressing mice during the testing session. Solid lines indicate the mean value and shadowed areas indicate the SEM. The black bar indicates the presence of the CS from 0 to 3 s. *p < 0.05, two-way RM ANOVA with post hoc Bonferroni multiple pairwise comparisons; SEM, standard error of the mean. (**h–i**) Schematic diagrams of chemogenetic CCK inhibition in the EC. Cre-dependent hM4Di was expressed in CCK-Cre mice. After Cre-mediated recombination, CCK neurons in the EC were transfected with hM4Di. (**j**) Verification of viral expression in the bilateral EC (top panel) and the EC-LA projection (bottom left panel). A magnified image of the EC-LA projection is shown in the bottom right inset. (**k–l**) Freezing percentages before (**k**) and after (**l**) the CS during the testing session in mice treated with CLZ or vehicle (VEH). *p < 0.05, two-way RM ANOVA with Bonferroni post hoc pairwise test; NS, not significant. (**m**) Freezing score plot of CLZ- and VEH-treated mice during the testing session. The black bar indicates the presence of the CS from 0 to 10 s. *p < 0.05, two-way RM ANOVA with post hoc Bonferroni multiple pairwise comparisons; SEM, standard error of the mean.

The online version of this article includes the following source data and figure supplement(s) for figure 4:

**Source data 1.** Summary of freezing percentage in mice with chemogenetic inhibition of the EC and EC-CCK neurons.

**Figure supplement 1.** Verification of chemogenetic suppression in the entorhinal cortex (EC) via in vivo electrophysiological recording.

= 0.026 < 0.05; pairwise comparison, CLZ vs. VEH at baseline, 12.9% ± 2.0% vs. 14.2% ± 2.0%, p = 0.644 > 0.05; CLZ vs. VEH post-training, 48.4% ± 5.8% vs. 27.1% ± 5.8%, p = 0.019 < 0.05; *Figure 4k*, two-way RM ANOVA, interaction not significant, F[1, 18] = 0.043, p = 0.838 > 0.05; pairwise comparison, CLZ vs. VEH at baseline, 10.2 % ± 1.9 vs. 9.4% ± 1.9%, p = 0.784 > 0.05; CLZ vs. VEH post-training, 18.0% ± 3.3% vs. 18.3% ± 3.3%, p = 0.949 > 0.05; *Videos 9–10*). These results mirror those observed in *Cck*[-/-] mice and suggest that trace fear memory formation relies on intact and functional CCK-positive neurons in the EC.

## CCK-positive neural projections are predominant in the EC-LA pathway

To further demonstrate that afferents to the amygdala originate from CCK-expressing neurons in the EC, we locally injected a Cre-dependent color-switching virus (AAV-CAG-DO-mCherry-DIO-EGFP) in the EC of CCK-Cre mice (N = 2; *Figure 5a–b*). With this combination, CCK-positive neurons express EGFP, and CCK-negative neurons express mCherry (*Saunders et al., 2012*). We found that EGFP+ (i.e., CCK+) neurons made up a slightly higher proportion of labeled neurons than mCherry+ (i.e., CCK–) neurons (*Figure 5c–d*, EGFP vs. mCherry, 58.9% ± 4.8% vs. 38.6% ± 5.0%, one-way RM ANOVA, Wilks' lambda = 0.58, F[1,6] = 4.34, p = 0.0822 > 0.05). Interestingly, we found that CCK + neural projections from the EC to the LA were densely labeled with EGFP, whereas mCherry labeling of CCK– projections was dramatically weaker. Quantitative analysis revealed that the projection intensity of the EC[CCK+] was threefold higher than the EC[CCK–] (35.6% ± 9.5%). In other words, CCK-positive afferents constituted approximately 75% of total afferents from the EC to the LA (*Figure 5e–f*).

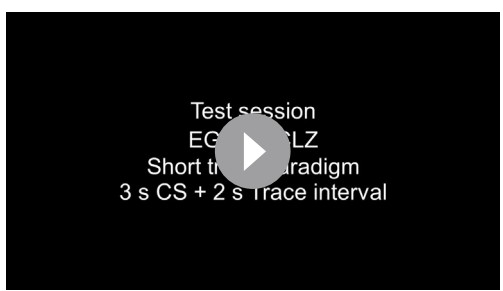

**Video 7.** Freezing response to the conditioned stimulus (CS) of EGFP-expressing mice treated with clozapine (CLZ) in the test session after short-trace fear conditioning paradigm, related to Figure 4e–f. EGFP mice showed significant freezing response to the CS after training.

https://elifesciences.org/articles/69333/figures#video7

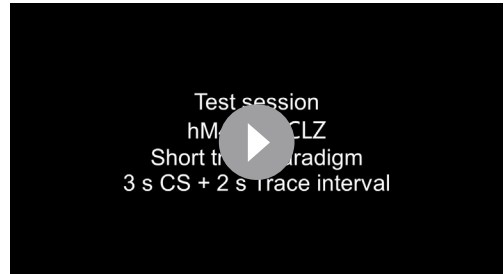

**Video 8.** Freezing response to the conditioned stimulus (CS) of hM4Di-expressing mice treated with clozapine (CLZ) in the test session after short-trace fear conditioning paradigm, related to Figure 4e–f. hM4Di mice showed impaired freezing response to the CS after training.

https://elifesciences.org/articles/69333/figures#video8

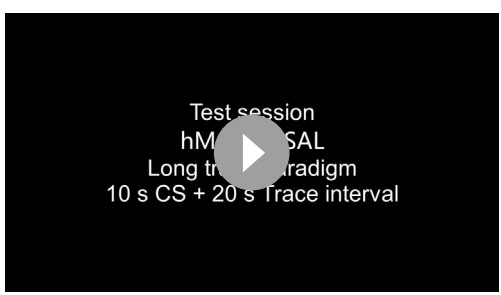

**Video 9.** Freezing response to the conditioned stimulus (CS) of hM4Di-expressing cholecystokinin (CCK)-Cre mice treated with vehicle in test session after long-trace fear conditioning paradigm, related to Figure 4k–l. Vehicle-treated mice showed significant freezing response to the CS after training.
https://elifesciences.org/articles/69333/figures#video9

To determine if the fluorescent reporter proteins interfered with projection strength, we inverted the color combination by combining two AAVs: AAV-hSyn-DIO-mCherry and AAV-EF1α-FAS-EGFP (*Saunders et al., 2012*). These Cre-dependent AAVs were injected into the EC of CCK-Cre mice. In CCK-Cre mice, AAV-hSyn-DIO-mCherry induces Cre-ON mCherry expression in CCK+ neurons, and AAV-EF1α-FAS-EGFP induces Cre-OFF EGFP expression in CCK– neurons (*Figure 5g–h*). With the mixed AAVs, we labeled approximately 50% CCK– EGFP+ neurons, 41% CCK+ mCherry + neurons, and 8.9% double-positive neurons (*Figure 5i–j*). The higher percentage of double-positive neurons present in this system indicates a higher probability of off-target effects compared to the previous color-switching AAV (8.9% ± 2.7% vs. 2.5% ± 1.1%). Consistent with the previous color-switching AAV, we observed that CCK+ (mCherry+) projections were predominant. Specifically, the intensity of the EC$^{CCK+}$ was approximately fourfold higher than the EC$^{CCK-}$ (24.0% ± 5.6%). Altogether, our results suggest that the EC$^{CCK+}$ is the predominant subpopulation of projections, and that these projections are of functional significance in the EC-LA pathway.

## CCK-positive neural projections from the EC to the LA enable neural plasticity

Furthermore, we asked whether CCK-positive projections from the EC modulate neural plasticity in the LA. First, we expressed a Cre-dependent high-frequency-responsive channelrhodopsin (ChR2) variant E123T (ChETA) under control of the universal EF1α promoter in CCK-Cre mice (*Figure 6a*). Then, we inserted optic fibers targeting the LA to illuminate EC$^{CCK+}$ projections and electrodes to conduct in vivo electrophysiological recording as before (*Figure 6b*). Post hoc anatomical analysis confirmed the distribution of ChETA in the EC-LA axon terminals (*Figure 6c*). Terminals of these CCK-positive projections were colocalized with CCKBR in the LA (*Figure 6d*), implying that CCK-positive projections from the EC may innervate with CCKBR in the LA. Finally, we recorded AEP and visual evoked potential (VEP) in the LA of anesthetized mice (*Figure 6e–g*). Although AEP and VEP had similar waveforms, the latency of AEP was much shorter than VEP (*Figure 6e–f*, peak latency: 38.9 ± 3.2 ms for AEP, N = 13, vs. 89.5 ± 3.1 ms for VEP, N = 11, two-sample t-test, t (22) = 11.376, p = 1.1E-10 < 0.001). This observation implies that the auditory and visual signal transmission pathway to the LA has different features. We applied high-frequency laser stimulation (HFLS, *Figure 6h*) of the EC-LA axons before the AS

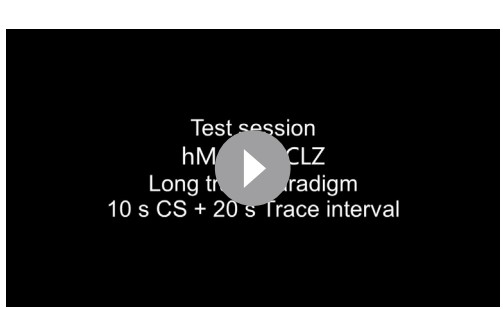

**Video 10.** Freezing response to the conditioned stimulus (CS) of hM4Di-expressing cholecystokinin (CCK)-Cre mice treated with clozapine (CLZ) in test session after long-trace fear conditioning paradigm, related to Figure 4k–l. CLZ-treated mice showed impaired freezing response to the CS after training.
https://elifesciences.org/articles/69333/figures#video10

to trigger AEP-LTP in the LA. After induction, the AEP slope in the ChETA-expressing group (n = 10) increased significantly, whereas the VEP slope did not change (*Figure 6i–j*, two-way RM ANOVA, significant interaction, F[1,9] = 14.46, p = 0.004 < 0.01; pairwise comparison, AEP before vs. after pairing, 97.8% ± 5.5% vs. 187.6% ± 15.6%; 95% CI, [85.3%, 110.3%] vs. [152.4%, 222.8]; p = 0.000258 < 0.001; VEP before vs. after pairing, 96.3% ± 4.9% vs. 120.7% ± 9.1%; 95% CI, [85.2%, 107.3%] vs. [100.1%, 141.3%], p = 0.091 > 0.05). Additionally, we injected a non-opsin expressing control AAV (AAV- EF1α-DIO-EYFP, n = 22) and the AEP-LTP was not induced with the same protocol (two-way RM ANOVA between CHETA and EYFP, F[1,30] = 46.65, p = 1.41E-7 < 0.001; pairwise comparison, before vs. after pairing in

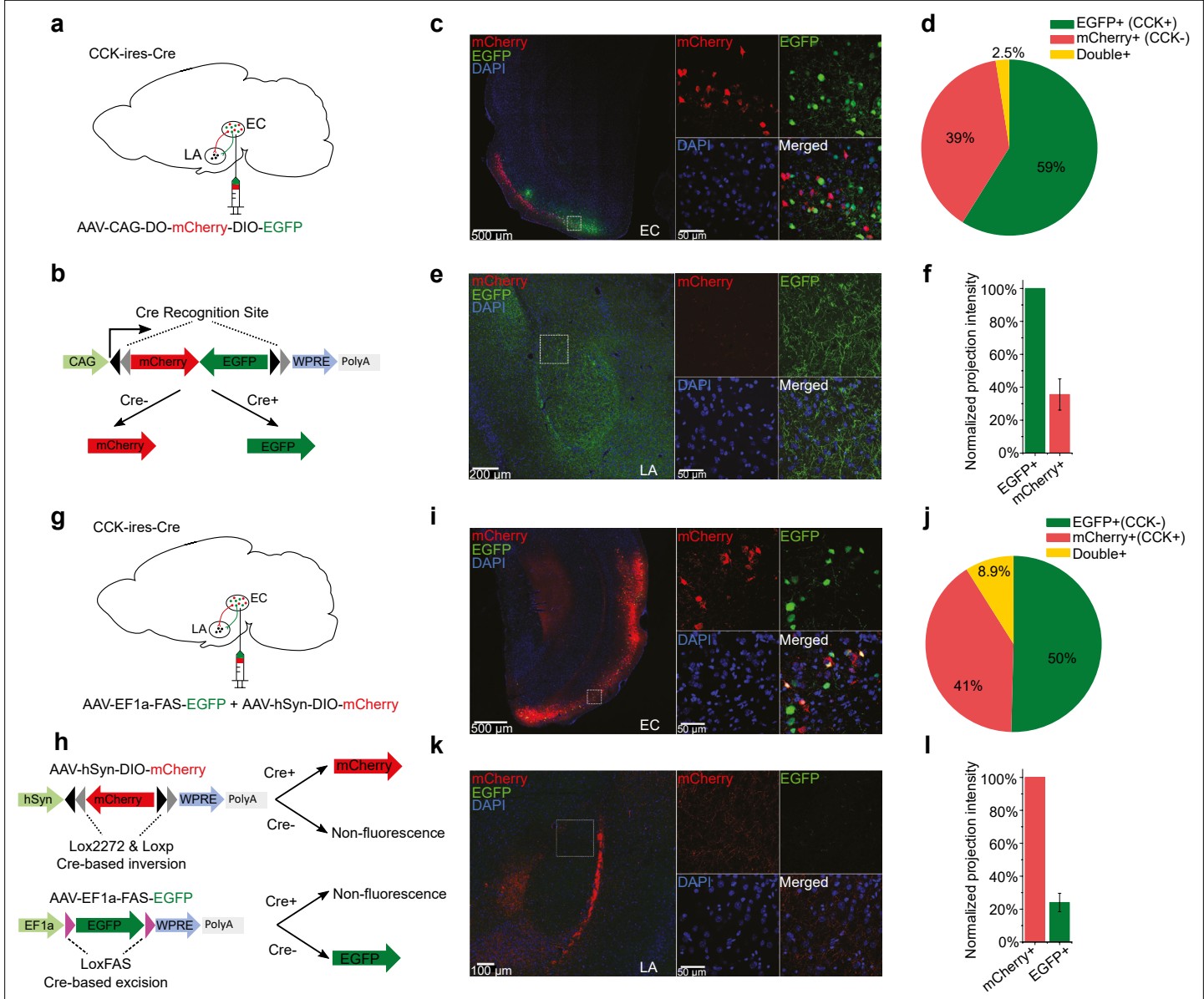

**Figure 5.** Cholecystokinin (CCK)-expressing projections predominate in the entorhinal cortex (EC)-lateral nuclei of the amygdala (LA) pathway. (**a–b**) Schematic diagram of Cre-dependent color-switch labeling in the EC-LA pathway. AAV-CAG-DO-mCherry-DIO-EGFP was injected in the EC. Using this labeling scheme, EGFP is expressed in CCK+ neurons, and mCherry is expressed in CCK– neurons. (**c–d**) Visualization (**c**) and quantification (**d**) of viral expression in the EC. Representative immunofluorescent images in the EC 7 weeks after viral injection (**c**). Scale bar = 500 μm (left). Magnified images are shown in insets on the right. Scale bar = 50 μm. Percentages of EGFP+ (CCK+), mCherry+ (CCK–), and double-positive neurons (**d**). No statistical differences were observed. p = 0.08; one-way RM ANOVA, repeated-measures analysis of variance. (**e–f**) Visualization (**e**) and quantification (**f**) of EGFP-expressing (CCK+) and mCherry-expressing (CCK–) afferents in the amygdala stemming from the EC. The fluorescent intensity of neuronal projections was normalized to the EGFP+ signal, which was approximately threefold stronger than the mCherry+ signal (35.6% ± 9.5%). (**g–h**) Schematic diagram of Cre-dependent color-switch labeling in the EC-LA pathway. A mixture of AAV-hSyn-DIO-mCherry and AAV-EF1α-FAS-EGFP was injected into the EC. Using this labeling scheme, mCherry is expressed in CCK+ neurons, and EGFP is expressed in CCK– neurons. (**i–j**) Visualization (**i**) and quantification (**j**) of viral expression in the EC. Representative immunofluorescent images in the EC 7 weeks after viral injection (**c**). Scale bar = 500 μm (left). Magnified images are shown in insets on the right. Scale bar = 50 μm. Percentages of mCherry+ (CCK+), EGFP+ (CCK–), and double-positive neurons (**j**). No statistical differences were observed. p = 0.55; one-way RM ANOVA; Wilks' lambda = 0.94; F(1,6) = 0.39. (**k–l**) Visualization (**k**) and quantification (**l**) of EGFP-expressing (CCK+) and mCherry-expressing (CCK–) afferents in the amygdala stemming from the EC. The fluorescent intensity of neuronal projections was normalized to the mCherry+ signal, which was approximately fourfold stronger than the EGFP+ signal (24.0% ± 5.6%).

The online version of this article includes the following source data for figure 5:

**Source data 1.** Quantification of viral expression and projection strength.

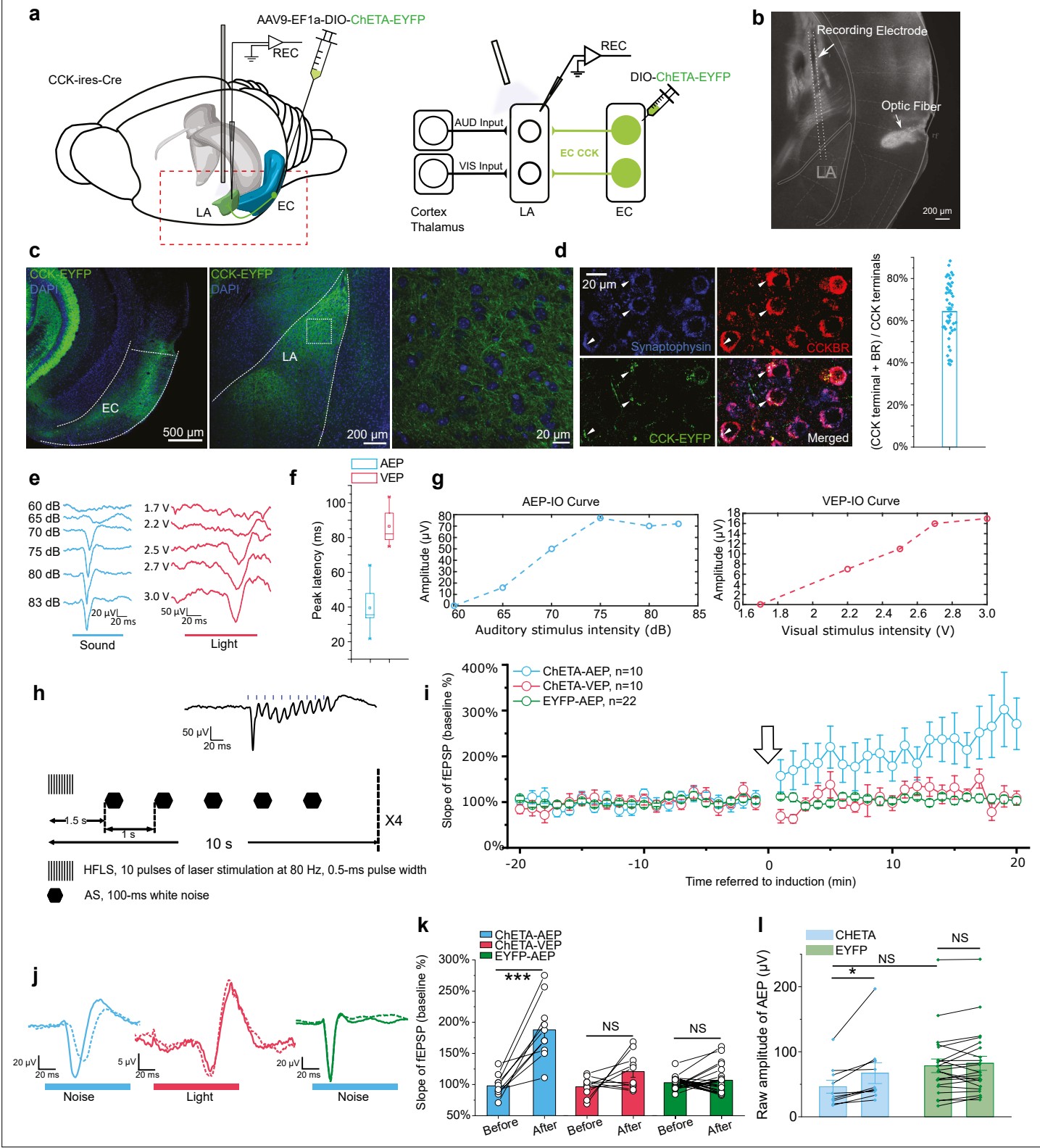

**Figure 6.** High-frequency activation of the EC^CCK+ pathway induces long-term potentiation (LTP) of auditory evoked potential (AEP) in the lateral nuclei of the amygdala (LA). (**a**) Schematic diagram of the experiment. The Cre-dependent high-frequency-responsive opsin ChETA was expressed in the EC of cholecystokinin (CCK)-Cre mice. Electrodes were inserted into the LA, and blue light was used to illuminate the recording area. The red rectangle in the left panel is magnified in the right panel to illustrate the neural pathways that are recruited during recording. AUD, auditory stimulus; VIS, visual stimulus; LA, lateral amygdala; EC, entorhinal cortex; REC, recording. (**b**) Post hoc verification of the electrode tracks and optic fiber placement. (**c**)

*Figure 6 continued*

Post hoc verification of viral expression in the EC (left) and in CCK-positive projections in the LA (middle). A magnified image is shown in the right panel and corresponds to the boxed area of the middle panel. (**d**) Co-immunofluorescent staining of the CCK-positive fiber (EYFP), the axon terminal (synaptophysin), and CCK B receptor (CCKBR) in the LA. The white arrowhead indicates a triple-positive neural terminal. Quantification of the CCK and CCKBR double-positive neural terminals out of all CCK-positive terminals (right). (**e**) Representative traces of AEP and visual evoked potential (VEP) at different sound and light intensities. (**f**) AEP and VEP peak latency. (**g**) Representative input/ouput (IO) curves for AEP (left) and VEP (right). (**h**) Detailed pairing protocol to induce LTP. Representative averaged fEPSP trace evoked by HFLS is shown in the inset. HFLS, high-frequency laser stimulation; AS, auditory stimulation. (**i**) Time course plot of the normalized slope of AEP and VEP during LTP. The arrow indicates the application of LTP induction. (**j**) Representative traces of averaged AEP/VEP before (–10 to 0 min, dotted line) and after (10–20 min, solid line) induction from the three groups. (**k**) The average normalized slopes 10 min before pairing (–10 to 0 min, before) and 10 min after pairing (10–20 min, after) in the three groups. ***$p < 0.001$, NS, not significant; two-way RM ANOVA with Bonferroni post hoc pairwise comparison. (**l**) The raw amplitude before (–10 to 0 min) and after (10–20 min) pairing in CHETA and EYFP groups. *$p < 0.05$, NS, not significant; two-way RM ANOVA with Bonferroni post hoc pairwise comparison.

The online version of this article includes the following source data for figure 6:

**Source data 1.** AEP-LTP induction with HFLS on CCK+ projection from the EC to the LA.

the EYFP group, 102.9% ± 2.7% vs. 106.7% ± 7.0%; 95% CI, [97.3%, 108.5%] vs. [92.4%, 120.9%]; p = 0.591 > 0.05, *Figure 6h–i*) These results suggest that high-frequency activation of $EC^{CCK+}$ switches the AEP-LTP in the LA.

In the next experiment, we examined the possibility of other neuroactive molecules co-released with CCK and contributing to HFLS-induced AEP-LTP. We adopted an RNA interference technique to knock down the *Cck* expression in the EC specifically. We accomplished this by injecting a Cre-dependent AAV cassette carrying a ChR2 variant (E123T/T159C) and a short hairpin RNA (shRNA) targeting *Cck* (anti-*Cck*) or a nonsense sequence (anti-Scramble) into the EC of CCK-Cre mice (*Figure 7a–c*). The knockdown efficiency on *Cck* expression was quantitatively verified by real-time PCR (*Figure 7d*). Meanwhile, we injected this virus in WT mice and found ChR2 was not expressed in the injected area, indicating a reliable Cre dependency of this AAV (*Figure 7—figure supplement 1*). The inclusion of laser-responsive ChR2 allowed us to induce the above AEP-LTP by specifically stimulating the $EC^{CCK+}$ pathway. We applied our HFLS pairing protocol in these mice and found that AEP-LTP could not be induced in the anti-*Cck* group but could be successfully induced in the anti-Scramble group (*Figure 7e–h*, two-way RM ANOVA, significant interaction, F[1,31] = 14.94, p = 0.00053 < 0.001; pairwise comparison, before vs. after pairing in the anti-*Cck* group, 101.5% ± 2.8% vs. 98.0% ± 6.5%; 95% CI, [95.7%, 107.4%] vs. [84.6%, 111.3%]; p = 0.594 > 0.05; before vs. after pairing in the anti-Scramble group, 103.0% ± 3.3% vs. 138.8% ± 7.6%; 95% CI, [96.2%, 109.8%] vs. [123.3%, 154.4%]; p = 0.000062 < 0.001). This observation implies that CCK alone is responsible for HFLS-induced AEP-LTP.

## CCK-positive neural projections from the EC to the LA specifically modulate the formation of trace but not delay fear memory

We employed optogenetics to dissect the real-time behavioral dependency of the trace fear memory formation on the $EC^{CCK+}$ pathway. We expressed a red-shifted inhibitory opsin Jaws (AAV8-hSyn-FLEX-Jaws-GFP) (*Chuong et al., 2014*) or mCherry control (AAV8-hSyn-DIO-mCherry) in the EC of CCK-Cre mice. We also implanted optical fibers targeting the bilateral LA in these mice and then subjected them to the long-trace fear conditioning (*Figure 8a–e*). During trace fear conditioning, $EC^{CCK+}$ were stimulated by a 635 nm red laser at a frequency of 5 Hz (i.e., 100 ms illumination +100 ms interval) through the optic fibers for the duration of the CS and trace interval, as indicated in *Figure 8c*. Freezing percentage to the CS was measured before (baseline) and after (post-training) this long-trace fear conditioning (*Figure 8e*). We found that mice expressed Jaws (Exp, N = 8/3 cages) had a prominent lower freezing percentage than mice expressed mCherry control (Ctrl, N = 9/3 cages), while in baseline session, there is no statistical difference between these groups (*Figure 8e*, two-way RM ANOVA, significant interaction, F[1,15] = 5.59, p = 0.032 < 0.05; in baseline session, Exp vs. Ctrl, 7.8 ± 2.1 % vs. 11.6 ± 2.0%; 95% CI, [3.4%, 12.2%] vs. [7.4%, 15.7%]; p = 0.208 > 0.05; in post-training session, Exp vs. Ctrl, 33.3% ± 5.3 % vs. 51.9% ± 5.0 %; 95% CI, [22.1%, 44.5%] vs. [41.4%, 62.5%]; p = 0.021 < 0.05; *Videos 11–12*). Also, we quantified the freezing percentage before the CS presentation in the baseline and post-training session to evaluate the basal freezing level without the CS (*Figure 8d*). We found no difference between the two groups (two-way RM ANOVA with Bonferroni pairwise comparison). From the freezing score plot in test day

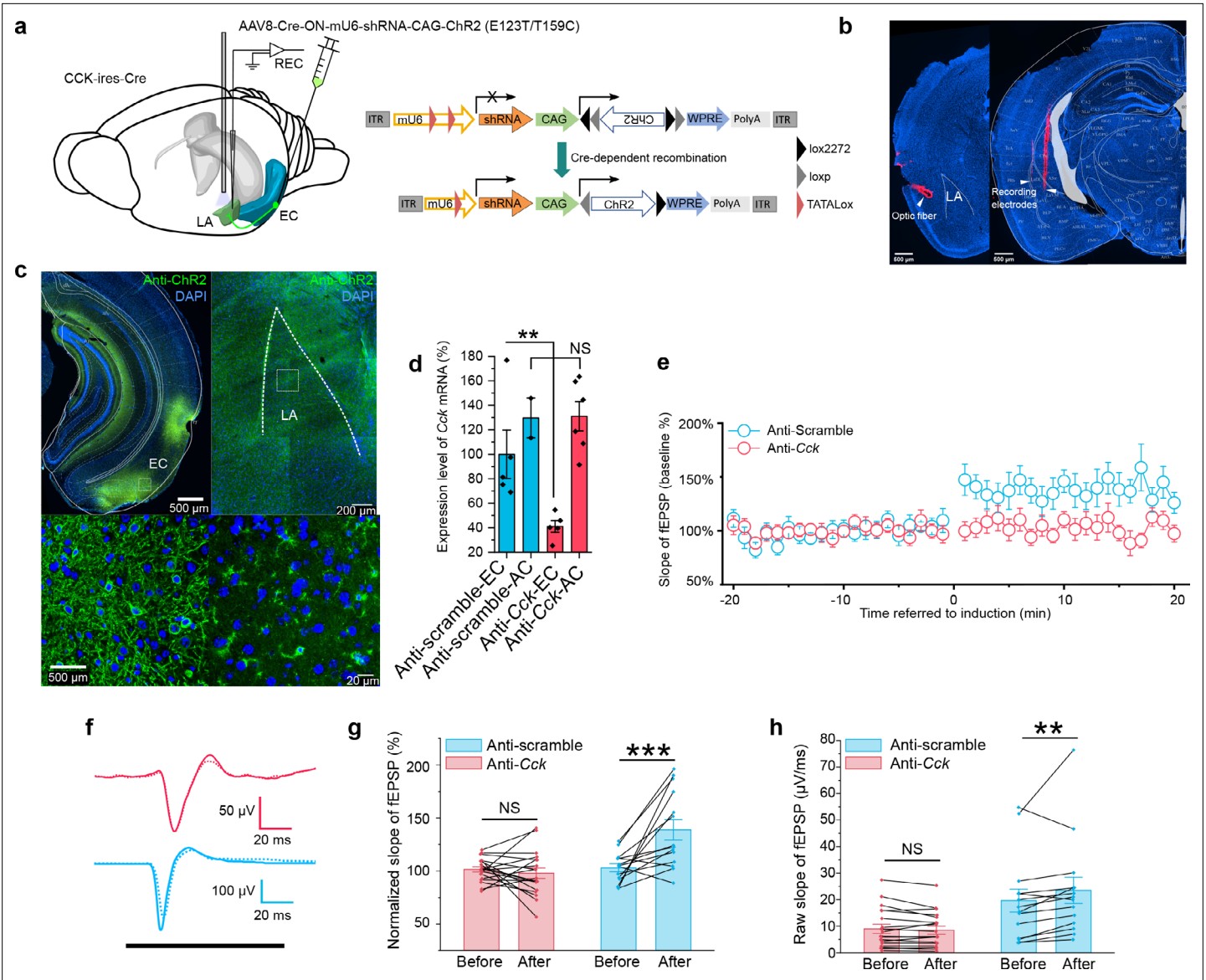

**Figure 7.** In vivo knockdown of *Cck* expression blocks auditory evoked potential (AEP)-long-term potentiation (LTP) induction in the LA. (**a**) Schematic diagram of the experiment. CCK-Cre mice were injected in the EC with an AAV expressing short hairpin RNA (shRNA) (anti-*Cck* or anti-Scramble) and ChR2. In vivo recording was conducted in the LA (left). After Cre-mediated recombination, EC-CCK neurons were transfected with shRNA targeting *Cck* (anti-*Cck*) or nonsense sequence (anti-Scramble) as well as the excitatory opsin ChR2 variant E123T/T159C (right). AAV, adeno-associated virus; EC, entorhinal cortex; LA, lateral amygdala; REC, recording; ITR, inverted terminal repeat; mU6, mouse U6 promoter; CAG, CMV enhancer, chicken β-actin promoter; WPRE, woodchuck hepatitis virus (WHP) posttranscriptional regulatory element. (**b**) Post hoc verification of the electrode tracks and optic fiber. (**c**) Post hoc immunofluorescent staining targeting ChR2 in the EC (left) as well as in the CCK-positive projections distributed in the LA (right). Magnified images are shown in the bottom insets. (**d**) Quantification of the expression of *Cck* mRNA in CCK-Cre mice injected with anti-*Cck* or anti-Scramble shRNA in the EC. Samples with extremely low RNA concentration (<26.7 ng/μL) were discarded. **p < 0.01, NS, not significant, two-way RM ANOVA with post hoc Bonferroni pairwise comparison. EC in anti-scramble group, N = 5; AC in anti-scramble group, N = 2; EC in anti-*Cck* group, N = 5; AC in anti-*Cck* group, N = 6. (**e**) Time course plot of the normalized AEP slope before and after pairing in mice expressing anti-*Cck* (n = 19) or anti-Scramble (n = 14) shRNA. (**f**) Representative traces of the averaged AEP before (–10 to 0 min, dotted line) and after (10–20 min, solid line) induction in the two groups. Anti-Scramble is indicated in blue, and anti-*Cck* is indicated in red. (**g**) The average normalized slopes 10 min before pairing (–10 to 0 min, before) and 10 min after pairing (10–20 min, after) in the two groups. ***p < 0.001, two-way RM ANOVA with Bonferroni post hoc pairwise test; RM ANOVA, repeated-measures analysis of variance; NS, not significant; fEPSP, field excitatory postsynaptic potential. (**h**) The average raw slopes 10 min before pairing (–10 to 0 min, before) and 10 min after pairing (10–20 min, after) in the two groups. **p < 0.01, two-way RM ANOVA with Bonferroni post hoc pairwise test.

The online version of this article includes the following source data and figure supplement(s) for figure 7:

*Figure 7 continued on next page*

*Figure 7 continued*

**Source data 1.** AEP-LTP induction in mice with knockdown of *Cck* expression in the EC.

**Figure supplement 1.** Verification Cre dependency of adeno-associated virus (AAV) carrying short hairpin RNA (shRNA) and ChR2.

(*Figure 8c*), we found the control group had a higher freezing score than the experimental group after the CS presentation (two-way RM ANOVA with a Huynh Feldt correction, F[23.6, 354.3] = 0.971, p = 0.503 > 0.05; Bonferroni multiple pairwise comparisons between two groups in each time point, *p = 0.048, 0.016 < 0.05 at time point 10–12 and 18–20 s referred to the onset of CS, respectively). The results indicate that opto-inhibition of the $EC^{CCK+}$ during the training session of fear conditioning can impair the formation of long-trace fear memory.

To test the specificity of the CCK pathway from EC to LA, we applied a long-delay fear conditioning paradigm, in which CS was 30-s-long to cover the whole trace interval and co-terminated with the US (*Figure 8f*). We did the same optogenetic manipulation on CCK-positive terminals in a new batch of mice (Exp, N = 11/3 cages; Ctrl, N = 10/3 cages). Interestingly, we found that after this long-delay conditioning, both groups of mice can obtain a high and similar freezing level in response to the CS presentation (*Figure 8h*, two-way RM ANOVA, interaction not significant, F[1, 19] = 1.12, p = 0.304 > 0.05; in baseline session, Exp vs. Ctrl, 8.6 ± 2.3 % vs. 7.2 ± 2.4%; 95% CI, [3.8%, 13.3%] vs. [2.2%, 12.2%]; p = 0.676 > 0.05; in post-training session, Exp vs. Ctrl, 80.0 ± 3.6 % vs. 72.2 ± 3.7%; 95% CI, [72.5%, 87.5%] vs. [64.3%, 80.0%]; p = 0.145 > 0.05; *Videos 13–14*). From the freezing score plot on the test day (*Figure 8f*), we observed a similar response curve to the CS, with some time points, the experimental group had a higher freezing score than the control group (two-way RM ANOVA with a Huynh Feldt correction, F[22.0, 418.8] = 1.56, p = 0.051 > 0.05; Bonferroni multiple pairwise comparisons between two groups in each time point, *p = 0.026, 0.003 < 0.05 at time point 24–26 and 26–28 s referred to the onset of CS, respectively).

We also test the effect of real-time optogenetic inhibition on $EC^{CCK+}$ in the short-trace fear conditioning in a head-fixed setup (*Figure 8—figure supplement 1*). We expressed the inhibitory opsin eNpHR3.0 (AAV-EF1α-DIO-eNpHR3.0-mCherry) or GFP control (AAV-hSyn-FLEX-GFP) in the EC of CCK-Cre mice. Same as above, optic fibers were implanted to target bilateral LA in these mice. During the short-trace fear conditioning, mice were positioned in a head-fixed setup on a movable surface, and an electrical tail shock was given as the US. $EC^{CCK+}$ was inhibited by a 561 nm laser illumination at a frequency of 5 Hz (i.e., 100 ms illumination +100 ms interval) for the duration of the CS and trace interval, as indicated in *Figure 8—figure supplement 1a*. After administration of the US, we most commonly observed flight (running). Interestingly, we found that after a few training trials, some GFP control mice (3/6 animals, data not shown) began running before the US was given, suggesting that GFP mice associate the CS with the US and make predictions in subsequent training trials (*Video 15*). In contrast, we observe much fewer conditioned defensive responses in the eNpHR group throughout the training process (1/8 animals and 2/40 observed training trials, data not shown, *Video 16*). Additionally, we recorded the freezing percentages in response to the CS before and after head-fixed fear conditioning (*Figure 8—figure supplement 1c–d*). We found that mice in the eNpHR group showed impaired freezing percentages post-training compared to mice in the GFP group (*Figure 8—figure supplement 1d*, two-way RM ANOVA, significant interaction, F[1,12] = 19.20, p = 8.93E-4 < 0.001; pairwise comparison, GFP vs. eNpHR post-training, 39.1% ± 3.7% vs. 12.2% ± 3.2%; 95% CI, [31.3%, 46.8%] vs. [5.6%, 18.9%]; p = 8.39E-4 < 0.001; *Videos 17–18*). We did not observe any differences between the two groups at baseline (*Figure 8—figure supplement 1d*, pairwise comparison, GFP vs. eNpHR at baseline, 12.7% ± 2.8% vs. 16.1% ± 2.5%; 95% CI, [6.5%, 18.9%] vs. [10.7%, 21.4%]; p = 0.389 > 0.05) or prior to the CS (*Figure 8—figure supplement 1c*, two-way RM ANOVA, interaction not significant, F[1, 12] = 0.67, p = 0.43 > 0.05; pairwise comparison, GFP vs. eNpHR post-training, 19.3% ± 5.4% vs. 17.8% ± 4.7%; 95% CI, [7.5%, 31.1%] vs. [7.6%, 28.0%]; p = 0.835 > 0.05). Altogether, our results suggest that short-trace fear memory formation is also disturbed by real-time inhibition of the $EC^{CCK+}$ pathway.

Collectively, with the real-time opto-inhibition on CCK projections from the EC to the LA, we found the specific involvement of the $EC^{CCK+}$ in the trace fear memory formation.

In summary, the release of the neuropeptide CCK from the EC neurons switches neural plasticity in the LA and facilitates the formation of trace fear memory. Dysfunction in any part of this pathway

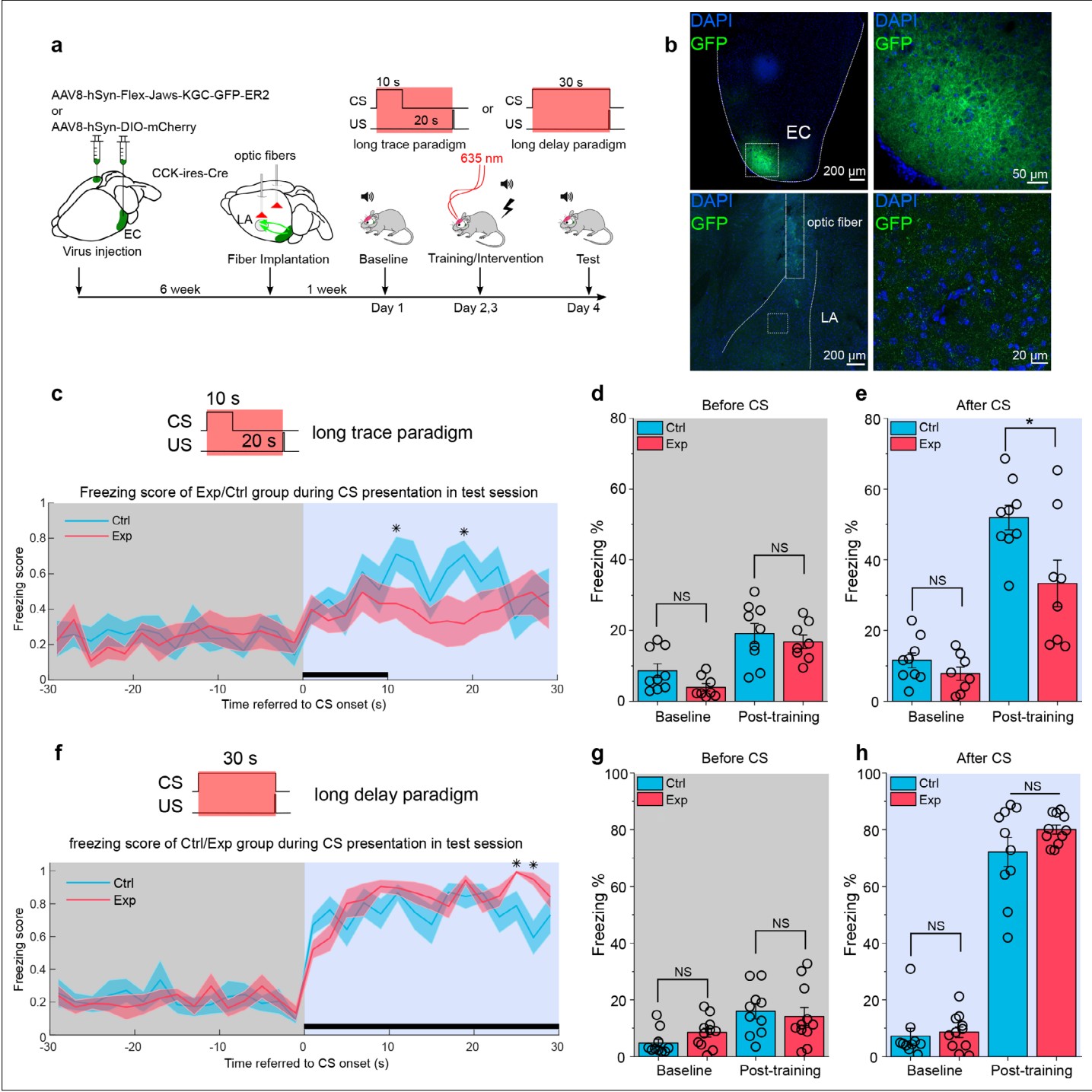

**Figure 8.** Real-time inhibition of the EC$^{CCK+}$ pathway impairs long-trace but not long-delay fear memory formation. (**a**) Schematic diagram of the experiment. The Cre-dependent inhibitory opsin Jaws or control was expressed in the EC of cholecystokinin (CCK)-Cre mice. Optic fibers were implanted targeted to the LA to illuminate and inhibit the CCK-positive projections from the EC to the LA during auditory-cued fear conditioning. Long-trace (10 s CS + 20 s trace +0.5 s US) and long-delay (30 s CS co-terminated with 0.5 s US) paradigms were used in current experiment. The inset at the top right shows the timing of 635 nm laser illumination. EC, entorhinal cortex; LA, lateral amygdala; CS, conditioned stimulus; US, unconditioned stimulus. (**b**) Post hoc verification of viral expression in the EC (top left) and of the optic fiber track in the LA (bottom left). Magnified images in the right panels show the transfected EC-CCK neurons (top right) and the CCK-positive EC-LA fibers (bottom right). (**c**) Freezing score across time during test session after long-trace fear conditioning. Mice expressed Jaws ('Exp', N = 8) had a relatively higher freezing score than mice expressed control virus ('Ctrl', N = 9). For all panels in this figure, *p < 0.05; NS, not significant; two-way RM ANOVA with Bonferroni pairwise comparison. (**d–e**) Freezing percentages before (**d**) and after (**e**) the CS in two groups of mice on pre-training day (baseline) and post-training day. (**g**) Freezing score across time

*Figure 8 continued on next page*

*Figure 8 continued*

during test session after long-delay fear conditioning. (**h–i**) Freezing percentages before (**h**) and after (**i**) the CS in two groups of mice on pre-training day (baseline) and post-training day.

The online version of this article includes the following source data and figure supplement(s) for figure 8:

**Source data 1.** Summary of freezing percentage in mice with opto-inhibition in long trace and long delay conditioning.

**Figure supplement 1.** Real-time inhibition of the EC<sup>CCK+</sup> pathway also impairs short-trace fear memory formation.

impairs the formation of trace fear memory in mice. These results extend our understanding of learning and memory formation and have important implications for fear-related mental disorders.

## Discussion

Here, we employed classical trace fear conditioning to test the formation of trace fear memory in *Cck*-/- and WT mice. We demonstrated that *Cck*-/- mice had impaired fear responses in both short- and long-trace fear conditioning. This behavioral defect was not caused by deficits in hearing and fear expression. Depleting CCK expression in mice impaired trace fear conditioning responses; this impairment was rescued by exogenous activation of CCKBR with its agonist CCK-4. Overall, our study suggests that trace fear memory formation and neural plasticity in the LA are dependent on a functional CCK network in the CNS.

Trace fear conditioning includes a gap between the CS and the US, distinguishing it from the simultaneous CS-US termination in delay fear conditioning. In trace fear conditioning, mice must retain information from the CS during the trace interval and associate it with the subsequent US. As a result, the learning process in trace fear conditioning is slower than in delay fear conditioning, and fear generalization is more pronounced. We previously reported that WT animals form CS-US associations after three training trials with minimal fear generalization in auditory-cued delay fear conditioning. In comparison, *Cck*-/- mice required nine training trials to achieve an equivalent freezing percentage to the CS (*Chen et al., 2019*). This result indicated the deficit of *Cck*-/- in the auditory-cued delay fear conditioning. We further demonstrated that *Cck*-/- mice also have difficulties in forming visually cued delay fear memory, as well as electrically cued trace fear memory in which an electrical pulse stimulus in the AC is paired with a foot shock (*Chen et al., 2019*; *Zhang et al., 2020*). Together, the results of our previous work and the present study indicate that the absence of the neuropeptide CCK has broad damaging effects on multiple forms of fear memory and is not limited to trace fear memory.

Fear conditioning can potentiate the signals of auditory-responsive units in the LA (*Quirk et al., 1995*) in a phenomenon referred to as LTP. As a result, many studies have identified LTP as a physiological hallmark of fear conditioning (*Blair et al., 2001*; *Maren, 2001*). Our study adopted in vivo recording to measure auditory-evoked field excitatory postsynaptic potential or AEP. We found no apparent abnormalities in AEP (such as amplitude or latency) in *Cck*-/- mice, suggesting that cortical

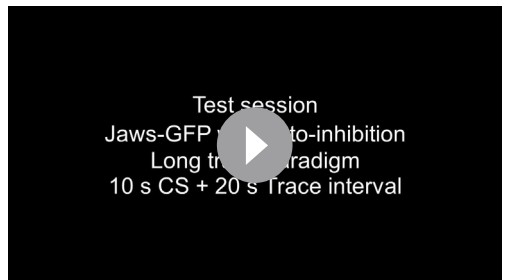

**Video 11.** Freezing response to the conditioned stimulus (CS) of Jaws-expressing mice (Exp) in test session after long-trace fear conditioning along with opto-inhibition, related to Figure 8c–e. Exp mice showed impaired freezing response to the CS after training.

https://elifesciences.org/articles/69333/figures#video11

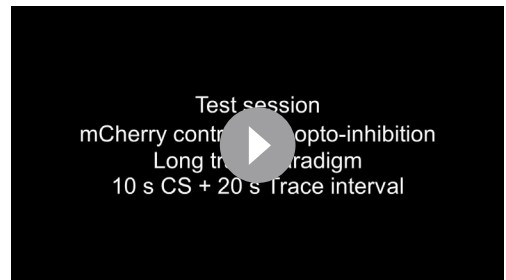

**Video 12.** Freezing response to the conditioned stimulus (CS) of mCherry-expressing mice (Ctrl) in test session after long-trace fear conditioning along with opto-inhibition, related to Figure 8c–e. Ctrl mice showed significant freezing response to the CS after training.

https://elifesciences.org/articles/69333/figures#video12

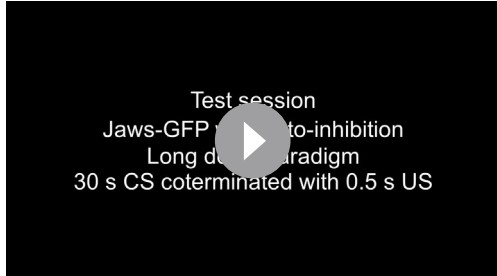

**Video 13.** Freezing response to the conditioned stimulus (CS) of Jaws-expressing mice (Exp) in test session after long-delay fear conditioning along with opto-inhibition, related to Figure 8f–h. Exp mice showed significant freezing response to the CS after training.
https://elifesciences.org/articles/69333/figures#video13

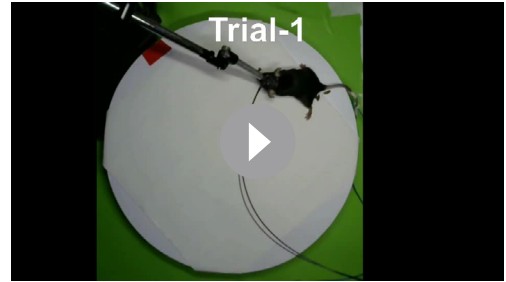

**Video 15.** Conditioned flight response to the conditioned stimulus (CS) of GFP-expressing cholecystokinin (CCK)-Cre mice illuminated with green light during short-trace fear conditioning. Mice showed apparent flight response in training trial 3.
https://elifesciences.org/articles/69333/figures#video15

and thalamic auditory inputs to the LA were functional. *Cck*[-/-] mice failed to induce AEP-LTP in the LA, strongly suggesting a deficiency in neural plasticity. We understand AEP-LTP induction is not equivalent to trace fear memory as it is not sufficient to trigger the expression of fear behaviors sometimes. LTP in the LA is maintained during fear extinction (*Kim and Cho, 2017*). Thus, LTP in the LA is necessary but not sufficient for fear memory formation.

In the present study, we found that silencing EC neurons with DREADD hM4Di impaired the formation of trace fear memory, consistent with several previous studies. Electrolytic lesion of the EC impairs trace eyeblink conditioning performance in mice (*Ryou et al., 2001*). Neurotoxic lesions in the EC impair the formation of trace fear memory but not that of delay fear memory formation (*Esclassan et al., 2009*). Although the hippocampus may involve in the trace fear memory formation (*Bangasser et al., 2006*), the EC is a promising regulatory region as it maintains sustained activity in response to stimuli (*Egorov et al., 2002*; *Fransén et al., 2006*). This sustained neuronal activity is thought to be the neural basis of 'holding' CS information during trace intervals to allow for CS-US association even after long-trace intervals (20 s in our study). This information 'holding' theory is consistent with neuroimaging reports on working memory in subjects who 'hold' stimuli for specific periods (*Nauer et al., 2015*).

Auditory responses have been previously found in the EC and its upstream circuit *Zhang et al., 2018*; however, these responses were limited to loud noise and did not involve the pure tone used in our behavioral paradigm. We reasoned that if the EC perceives and delivers the CS to downstream structures, then lesions in the EC would disturb the delay fear conditioning as well. Instead, previous studies have robustly demonstrated that EC lesions leave delay fear memory intact (*Esclassan et al., 2009*). The amygdala responds directly to the AS, and receives inputs from the AC, the MGB, and hippocampus. Thus, the EC is likely involved in the CS-US association more

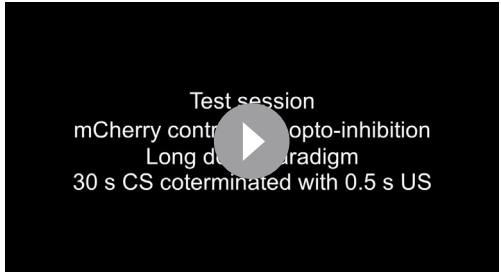

**Video 14.** Freezing response to the conditioned stimulus (CS) of mCherry-expressing mice (Ctrl) in test session after long-delay fear conditioning along with opto-inhibition, related to Figure 8f–h. Ctrl mice showed significant freezing response to the CS after training.
https://elifesciences.org/articles/69333/figures#video14

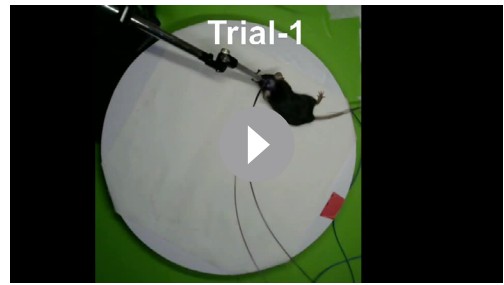

**Video 16.** Conditioned flight response to the conditioned stimulus (CS) of eNpHR-expressing cholecystokinin (CCK)-Cre mice illuminated with green light during short-trace fear conditioning. Flight response was blocked.
https://elifesciences.org/articles/69333/figures#video16

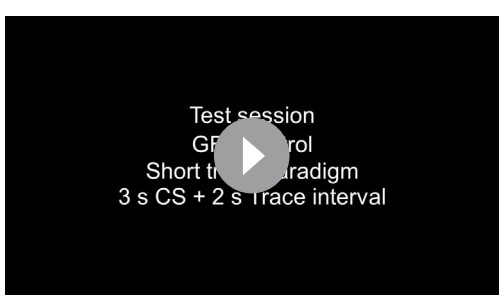

**Video 17.** Freezing response to the conditioned stimulus (CS) of GFP-expressing cholecystokinin (CCK)-Cre mice in the test session after short-trace fear conditioning paradigm, related to Figure 8—figure supplement 1c-d. GFP mice showed significant freezing response to CS after training.
https://elifesciences.org/articles/69333/figures#video17

complicated, and this mechanism requires further investigation. We speculate that this mechanism is probably similar to our previous finding in the sound-sound association (*Chen et al., 2019*) and visuo-auditory association (*Zhang et al., 2020*), which is neuropeptide-based hetero-synaptic modulation machinery.

With cell type-specific tracing systems, we demonstrated that the EC is an upstream brain region that projects CCK-positive afferents to the LA. These CCK-expressing EC neurons are primarily excitatory (*Figure 3*). Using anterograde Cre-dependent color switch labeling in the EC, we found that CCK-expressing neurons were the predominant source of EC-LA projections, implying that CCK is integral to EC-LA connection and communication. Cell type-specific chemogenetic inhibition of CCK-expressing neurons in the EC also impaired the formation of trace fear memory. However, we cannot exclude the possibility that CCK may originate in other brain regions and contribute to fear memory formation.

In a previous publication, we induced the release of CCK from terminals by HFLS on CCK-expressing fibers (*Chen et al., 2019*), which is consistent with the theory proposed several decades ago (*Hökfelt, 1991*). In the current study, we employed the same protocol to trigger the CCK released from CCK-positive terminals in the LA under in vivo preparation. We then presented the AS in the presence of this artificially released CCK neuropeptide. The auditory inputs from both the AC and the thalamus activated presynaptic axons via the canonical LA fear circuit (*Romanski and LeDoux, 1992*). In our study, the AS triggered postsynaptic neural firing. Therefore, our HFLS-mediated AEP-LTP induction protocol combines the released CCK with pre- and postsynaptic activation altogether in the LA, and this pairing leads to the potentiation of AEP in the LA.

We successfully excluded the contribution of substances co-released with CCK to the AEP-LTP induction, by blocking it after knocking down of *Cck* expression with shRNA. Our results that the inhibition of CCK-positive EC afferents to the LA impaired trace memory formation during the learning and response phases suggest that establishing the CS-US association during trace fear conditioning requires functional CCK-positive EC-LA projections.

Of note, we cannot underestimate the dependence of trace fear memory on contextual fear memory because some critical areas, include the hippocampus (*McEchron et al., 1998*) and the mPFC (*Gilmartin and Helmstetter, 2010*), contribute to both types of fear memory. EC bridges the hippocampus and the neocortex and is crucial for the integration of spatial information. The projections from the amygdala to the EC are suggested to participate in the contextual fear conditioning (*Wahlstrom et al., 2018*). Therefore, our unveiled CCK-positive EC-LA projections may also involve the formation of contextual fear.

In conclusion, we found that EC-LA projections modulate neuroplasticity in the LA and contribute to the formation of trace fear memory. The EC neurons release CCK in the LA, enabling hetero-synaptic neuroplasticity of the auditory inputs to the LA. Our findings add a novel insight into the participation of the neuropeptide CCK in the formation of the trace fear memory. As various mental disorders, including anxiety (*Davis, 1992*), depression (*Siegle et al., 2007*; *Shen et al., 2019*), and PTSD (*Shin et al., 2006*), are highly correlated with hyperactivation and dysfunction

**Video 18.** Freezing response to the conditioned stimulus (CS) of eNpHR-expressing cholecystokinin (CCK)-Cre mice in the test session after short-trace fear conditioning paradigm, related to Figure 8—figure supplement 1c-d. eNpHR mice showed impaired freezing response to CS after training.
https://elifesciences.org/articles/69333/figures#video18

of the amygdala and the fear memory circuitry, our finding supports CCK and its receptors as potential new targets for future therapeutic applications for these disorders.

# Materials and methods

## Key resources table

| Reagent type (species) or resource | Designation | Source or reference | Identifiers | Additional information |
|---|---|---|---|---|
| Antibody | Anti-CCKBR (Rabbit polyclonal) | Thermo Fisher Scientific | Cat# PA3-201, RRID:AB_10979062 | IF (1:1000) |
| Antibody | Anti-CCKBR (Mouse monoclonal) | Santa Cruz Biotechnology | Cat# sc-166690, RRID:AB_2070487 | IF (1:200) |
| Antibody | Anti-Synaptophysin (Mouse monoclonal) | Sigma-Aldrich | Cat# S5768, RRID:AB_477523 | IF (1:500) |
| Antibody | Anti-CamKII$\alpha$ (Rabbit monoclonal) | Abcam | Cat# Ab52476, RRID:AB_868641 | IF (1:500) |
| Antibody | Anti-GAD67 (Mouse monoclonal) | Millipore | Cat# MAB5406, RRID:AB_2278725 | IF (1:500) |
| Antibody | Anti-ChR2 (Mouse monoclonal) | American Research Products | Cat# 03–651180 | IF (1:2000) |
| Antibody | Anti-mouse IgG Alexa 647 (Donkey polyclonal) | Jackson ImmunoResearch Labs | Cat# 715-605-150, RRID:AB_2340862 | IF (1:500) |
| Antibody | Anti-rabbit IgG Alexa 647 (Donkey polyclonal) | Jackson ImmunoResearch Labs | Cat# 711-605-152, RRID:AB_2492288 | IF (1:500) |
| Antibody | Anti-mouse IgG DyLight 594 (Goat polyclonal) | Thermo Fisher Scientific | Cat# 35511, RRID:AB_1965950 | IF (1:500) |
| Antibody | Anti-mouse IgG Alexa 488 (Donkey polyclonal) | Jackson ImmunoResearch Labs | Cat# 715-545-150, RRID:AB_2340846 | IF (1:500) |
| Antibody | Anti-mouse IgG Alexa 594 (Goat polyclonal) | Jackson ImmunoResearch Labs | Cat# 111-585-144, RRID:AB_2307325 | IF (1:500) |
| Recombinant DNA reagent | AAV-Ef1$\alpha$-DIO-ChETA-EYFP | Addgene | RRID:Addgene_26968 | |
| Recombinant DNA reagent | AAV-EF1$\alpha$-DIO-EYFP | BrainVTA | N/A | N/A |
| Recombinant DNA reagent | AAV-hSyn-FLEX-GFP | BrainVTA | N/A | N/A |
| Recombinant DNA reagent | AAV-hSyn-hM4Di-EGFP | BrainVTA | N/A | N/A |
| Recombinant DNA reagent | AAV-hSyn-EGFP | Addgene | RRID:Addgene_105539 | N/A |
| Recombinant DNA reagent | AAV-hSyn-DIO-hM4D(Gi)-mCherry | Addgene | RRID:Addgene_44362 | N/A |
| Recombinant DNA reagent | AAV-hSyn-DIO-mCherry | Addgene | RRID:Addgene_50459 | N/A |
| Recombinant DNA reagent | AAV-EF1$\alpha$-DIO-eNpHR3.0-mCherry | BrainVTA | N/A | N/A |
| Recombinant DNA reagent | AAV-EF1$\alpha$-FAS-EGFP | Taitool | N/A | N/A |
| Recombinant DNA reagent | AAV-CAG-DO-mCherry-DIO-EGFP | This paper | N/A | AAV virus expressing Cre-On EGFP and Cre-Off mCherry |
| Recombinant DNA reagent | AAV8-Cre-ON-ChR2-anti*Cck* | This paper | N/A | AAV virus expressing Cre-dependent ChR2 and Cre-dependent shRNA targeting *Cck* |

*Continued on next page*

*Continued*

| Reagent type (species) or resource | Designation | Source or reference | Identifiers | Additional information |
|---|---|---|---|---|
| Recombinant DNA reagent | AAV8-Cre-ON-ChR2-antiScramble | This paper | N/A | AAV virus expressing Cre-dependent ChR2 and Cre-dependent shRNA targeting nonsense Scramble |
| Recombinant DNA reagent | retroAAV-hSyn-FLEX-jGcamp7s | Addgene | RRID:Addgene_104491 | N/A |
| Recombinant DNA reagent | AAV-hSyn-CCK2.0 | Vigene Bioscience, *Jing et al., 2019* | | Construct is from Prof. Yulong Li's Lab at Peking University |
| Recombinant DNA reagent | pAAV-CAG-Flex-tdTomato | Addgene | RRID:Addgene_28306 | N/A |
| Recombinant DNA reagent | PUC57-mU6 with TATALox | BGI, *Ventura et al., 2004* | | N/A |
| Recombinant DNA reagent | PUC57-CAG-DIO-ChR2(E123T/T159C)-Flag | Addgene | RRID:Addgene_35509; Addgene_101766 | N/A |
| Recombinant DNA reagent | pUC57-CAG-DIO-mCherry-EYFP (inverted) | Addgene | RRID:Addgene_34582; Addgene_98750 | N/A |
| Recombinant DNA reagent | AAV8-hSyn-FLEX-Jaws-GFP | UNC, *Chuong et al., 2014* | | N/A |
| Sequence-based reagent | *Cck* | BGI | shRNA Target | GACTCCCAGACCTAATGTTGC |
| Sequence-based reagent | Scramble | BGI | shRNA Target | GTTGGCTCCTAGCAGATCCTA |
| Sequence-based reagent | Primers for genotyping *Cck*−/− mice | BGI | PCR primers | 5'-ATGCAGGCAAATTTTGGTGT-3'; 5'-GAGCGGACACCCTTACCTTT-3'; 5'-GACTTCTGTGTGCGGGACTT-3 |
| Sequence-based reagent | *Gapdh* (Forward) | BGI | qPCR primer | 5'-AGGTCGGTGTGAACGGATTTG-3' |
| Sequence-based reagent | *Gapdh* (Reverse) | BGI | qPCR primer | 5'-TGTAGACCATGTAGTTGAGGTCA-3' |
| Sequence-based reagent | *Cck* (Forward) | BGI | qPCR primer | 5'-ATCTGTCCAGAGTGTGCAATGC-3' |
| Sequence-based reagent | *Cck* (Reverse) | BGI | qPCR primer | 5'-TGAGGGGCAGAAGGAAATCTCT-3' |
| Chemical compound, drug | Urethane | Sigma-Aldrich | Cat# U2500 | N/A |
| Chemical compound, drug | Pentobarbital | Alfasan International B.V. | | 20% Dorminal |
| Chemical compound, drug | CCK-4 | Abcam, Cambridge, UK | Cat# ab141328 | N/A |
| Chemical compound, drug | DiI Stain | Thermo Fisher Scientific | Cat# D282 | N/A |
| Chemical compound, drug | Clozapine | Sigma-Aldrich | Cat# C6305 | N/A |
| Peptide, recombinant proteins | Alexa Fluor 647-conjugated Cholera Toxin Subunit B | Thermo Fisher Scientific | Cat# C34778 | N/A |
| Genetic reagent (*Mus musculus*) | Mouse: C57BL/6 | The Laboratory Animal Services Centre, Chinese University of Hong Kong | N/A | N/A |

*Continued on next page*

*Continued*

| Reagent type (species) or resource | Designation | Source or reference | Identifiers | Additional information |
|---|---|---|---|---|
| Genetic reagent (*Mus musculus*) | Mouse: C57BL/6 | Laboratory Animal Research Unit, City University of Hong Kong | N/A | N/A |
| Genetic reagent (*Mus musculus*) | Mouse: CCK-ires-Cre | Jackson Laboratories | Stock# 012706 | N/A |
| Genetic reagent (*Mus musculus*) | Mouse: CCK-CreER | Jackson Laboratories | Stock# 012710 | N/A |
| Genetic reagent (*Mus musculus*) | Mouse: CCK-ABKO | Jackson Laboratories | Stock# 006365 | N/A |
| Genetic reagent (*Mus musculus*) | Mouse: CCK-BR KO | Jackson Laboratories | Stock# 006369 | N/A |
| Software, algorithm | Origin 2018 | OriginLab | https://www.originlab.com/2018 | N/A |
| Software, algorithm | Matlab R2020a | Mathworks | https://www.mathworks.com/products/new_products/release2020a.html | N/A |
| Software, algorithm | Fiji | *Schindelin et al., 2012* | https://imagej.net/Fiji | N/A |
| Software, algorithm | TDT OpenEX | Tucker-Davis Technologies | https://www.tdt.com/component/openex-software-suite/ | N/A |
| Software, algorithm | Photoshop CC | Adobe | https://www.adobe.com/products/photoshop.html | N/A |
| Software, algorithm | Excel | Microsoft | https://www.microsoft.com/en-us/microsoft-365/excel | N/A |
| Software, algorithm | Inkscape | N/A | https://inkscape.org/ | N/A |
| Software, algorithm | Offline Sorter | Plexon | https://plexon.com/products/offline-sorter/ | N/A |
| Software, algorithm | NeuroExplorer | Plexon | https://plexon.com/products/neuroexplorer/ | N/A |
| Software, algorithm | Bonsai | *Lopes et al., 2015* | https://bonsai-rx.org/ | N/A |
| Software, algorithm | CellProfiler | *McQuin et al., 2018* | https://cellprofiler.org/ | N/A |

## Animals

Adult male and female C57BL/6, *Cck*^-/- (CCK-CreER), and CCK-Cre (CCK-ires-Cre) mice were used in experiments. For behavioral experiments, only adult male mice were used. Mice were housed in a 12 hr shift of the reversed light-dark cycle and were provided food and water ad libitum. All behavioral experiments were conducted in the dark cycle. All experimental procedures were approved by the Animal Subjects Ethics Sub-Committee of the City University of Hong Kong.

For surgical procedures when doing virus injection and optic fiber implantation, mice were anesthetized with pentobarbital sodium (80 mg/kg, i.p., 20% Dorminal, Alfasan International B.V., Woerden, The Netherlands). For acute electrophysiological recording, mice were anesthetized with pentobarbital sodium (80 mg/kg, i.p.) or urethane sodium (2 g/kg, i.p., Sigma-Aldrich, St. Louis, MO). Both anesthetics were periodically supplemented during the experiment to maintain anesthesia. Mice were fixed in a stereotaxic device, and the scalp was incised. A local anesthetic (xylocaine, 2%) was applied

to the incision site for analgesia. After skull levelling, craniotomies were performed with varying parameters based on the region of the brain being accessed.

## Auditory and visual stimuli

AS, including pure tones and white noise, were digitally generated by a specialized auditory processor (RZ6 from Tucker-Davis Technologies [TDT], Alachua, FL). For behavioral experiments, AS were delivered via a free-field magnetic speaker (MF-1, TDT) mounted 60 cm above the animal. The sound intensity was adjusted by a condenser microphone (Center Technology, Taipei) to ~70 dB when it reached the animal. For in vivo recording, AS were delivered via a close-field speaker placed contralaterally to the recording side. The sound intensity that induced 50–70% of the maximum response was selected. Visual stimuli were generated by a direct current-driven torch bulb via the analog voltage output of the TDT workstation. Light intensity was roughly quantified as the value of the trigger voltage. For in vivo recording, the light intensity that induced 50–70% of the maximum response was selected.

## ABR recording

Mice were anesthetized with pentobarbital sodium (80 mg/kg, i.p.) and placed on a clean and warm blanket in a soundproof chamber. A free-field magnetic speaker (MF-1, TDT) was placed 10 cm away from the right ear of mice. Recording, reference, and ground needle electrodes (Spes Medica, Genova, Italy) were subcutaneously inserted below the forehead, right ear, and left ear, respectively. AS (wide spectrum clicks, 0.1 ms) were presented to the mouse with a decreasing level from 80 to 20 dB with an interval of 5 dB. For each level of click stimulus, total 512 times of presentations were given at a frequency of 21 Hz. ABR signals were collected via a specialized processor (RZ6, TDT) and digitalized with a bandpass filter from 100 Hz to 5 kHz. Stimuli generation and data processing were performed with software BioSigRZ (TDT).

## Trace fear conditioning

On the pre-conditioning day, each mouse was placed into the testing context (acrylic box with white wallpaper measuring 25 cm × 25 cm × 25 cm) for habituation and baseline recording. After 3 min of habituation, a CS (2.7 or 8.2 kHz pure tone, 70 dB SPL, 3 s for the short-trace paradigm and 10 s for the long-trace paradigm) was given three times within 20 min.

On conditioning day, the mouse was placed into the fear conditioning context (acrylic box with brown wallpaper measuring 18 cm wide ×18 cm long ×30 cm high and equipped with foot shock stainless steel grid floor). After 3 min of habituation, a CS-US pairing was given. In the short-trace interval paradigm, an US (0.5 mA foot shock, 0.5 s) was given 2 s after a 3-s-long CS. Three trials were given on each training day, and the interval between trials was 10–15 min. Totally two training days were given. The mouse was kept in the fear conditioning context for a 10 min consolidation period after the last training trial. In the long-trace interval paradigm, an US was given 20 s after a 10-s-long CS. Eight training trials were given each training day, and the interval between trials was 2–3 min. The mouse was kept in the fear conditioning context for a 5 min consolidation period after the last training trial. After training, each animal was kept in a temporary cage and returned to its home cage after all individuals finished training.

On post-conditioning day (test day), the mouse was placed into the testing context. After 3 min of habituation, a CS was presented to the animal twice with a 2-min-long interval between stimuli. Two minutes after the last trial, the animal was transferred to a temporary cage and returned to its home cage after all individuals in its cage finished testing.

All contexts were cleaned thoroughly with 75% ethanol after each individual session. All of the above procedures were conducted in a soundproof chamber, and all videos (baseline, training, and testing) were recorded with a webcam (Logitech C270) set in the ceiling of the chamber. Videos were analyzed with a custom program based on an open-source platform (*Lopes et al., 2015*) (https://bonsai-rx.org). Briefly, the centroid of the animal was extracted from the videos. By comparing the coordinates of the centroid frame by frame, we then calculated the distance moved between two frames. The instant velocity of the animal was calculated by dividing this distance by the time span between two adjacent frames. The freezing percentage was defined as the percentage of frames with an instant velocity lower than the threshold of all frames in an observed time window. We compared the output of this program to results observed by the naked eye. Finally, we selected 0.1 (pixel$^2$/s)

as the appropriate moving threshold to define freezing. Freezing score was defined as the binary value (0 or 1) of time frame with instant velocity higher (0, 'not freezing') or lower (1, 'freezing') than the threshold. For the freezing score plot shown in *Figures 1, 2 and 4*, freezing scores from all test sessions were averaged per second for data visualization.

## Electrophysiological recording in the LA and EC

Mice were subjected to the surgical procedures described above. Tracheotomy was conducted to facilitate breathing and to prevent asphyxia caused by tracheal secretions during the experiment. Craniotomy was performed 1.0–2.0 mm posterior and 3.0–4.0 mm lateral to the bregma to target the LA. Dura mater was partially opened using a metal hook made of a 29 G syringe needle. Tungsten recording electrodes (0.5–3.0 MΩ, FHC, Bowdoin, ME) were slowly inserted into the LA (approximately 3.5 mm from the brain surface). For laser stimulation experiments, another craniotomy was performed at the temporal lobe (1.0–2.0 mm posterior to the bregma) to expose the lateral rhinal vein. One optic fiber (200 μm diameter, 0.22 NA, Thorlabs, Newton, NJ) was inserted below the rhinal vein and forwarded till 1.0–1.5 mm from the surface. The angle of the optic fiber was approximately 75° from the vertical reference. Responses were recorded and passed to a pre-amplifier (PZ5, TDT) and an acquisition system (RZ5D, TDT). Signals were filtered for field potential or spikes with respective bandwidth ranges of 10–500 and 1–5000 Hz. All recordings were stored using TDT software (OpenEx, TDT). The maximum sound intensity was defined as the intensity that elicited a saturated AEP. The AEP baseline was recorded with 50% of the maximum sound intensity at a 5 s intertrial interval for 20 min. For high-frequency electrical stimulation experiments, we used ~ 70% of the maximum sound intensity and a 150 μA electrical stimulation current. For HFLS experiments, we used >10 mW laser power to ensure activation of transfected axons. After AEP-LTP induction, we recorded the AEP for another 20 min.

For recording in the EC, we applied the protocol from the Li I. Zhang Laboratory (*Zhang et al., 2018*). Craniotomy was performed at the juncture of the temporal, occipital, and interparietal bones and exposed the caudal rhinal vein and the transverse sinus (*Figure 4—figure supplement 1*). Electrodes were inserted approximately 1 mm below the dura mater.

All field potential data were extracted and processed in the MATLAB program, and all single-unit data were extracted from the TDT data tank to the Offline Sorter (Plexon) for spike sorting. Sorted data were forwarded to the Neuroexplorer (Plexon) for additional processing and visualization.

## Plasmid construction and AAV packaging

The sequence and cloning details of plasmid will be described elsewhere (Su et al., manuscript in preparation). In principle, we generated AAV vectors that allow Cre-controlled expression of shRNA and channelrhodopsin in neurons. For plasmid pAAV-Cre-ON-mU6-ShRNA-CAG-ChR2(E123T/T159C), shRNA was placed under the control of a mouse U6 (mU6) promoter inserted with a TATALox element (*Ventura et al., 2004*). CAG-DIO-ChR2(E123T/T159C) cassette was inserted following the mU6-TATALox-ShRNA cassette.

In brief, the pAAV backbone was recovered after digesting pAAV-CAG-Flex-tdTomato (Addgene 28306) with NdeI and HindIII. Fragment 1 (pUC57-Cre-ON-mU6-shRNA) was acquired by digesting pUC57-Cre-ON-mU6(TATALox) with HpaI and XhoI and then ligating it with annealed oligos that targets the coding sequence of *Cck* mRNA (Anti-*Cck*) or nonsense sequence (Anti-Scramble). Fragment 2 was acquired by digesting pUC57-CAG-DIO-ChR2(E123T/T159C-Flag) with XhoI and HindIII. Fragment 3 was acquired by digesting pUC57-CAG-DIO-mCherry-EYFP (inverted) with EcoRI and HindIII. pAAV backbone, Fragment 1 and Fragment 2, was ligated to make pAAV-Cre-ON-mU6-ShRNA-CAG-DIO-ChR2 (E123T/T159C)-Flag. pAAV backbone, Fragment 1 without shRNA, Fragment 3, was ligated to make pAAV-CAG-DO-mCherry-DIO-EYFP. DNA templates and shRNA oligos mentioned above were acquired from Addgene or synthesized from BGI (Shenzhen, China) and verified by sequencing.

For AAV packaging (*Xiong et al., 2015*), HEK293T cells were seeded into five dishes (15 cm, poly-D-lysine coated) for one viral preparation 1 day before transfection. Standard medium (DMEM, + 10% FBS and antibiotics) were used for HEK293T cells. For PEI transfection, mix 35 μg AAV8 helper plasmid, 35 μg AAV vector, 100 μg pHGTI-adenol, 510 μL of PEI (1 μg/mL, Sigma) with DMEM (without FBS or antibiotics) to final volume of 25 mL. Incubate this mixture at room temperature for 15 min. Meanwhile, replace the media in dishes with DMEM + 10% NuSerum (Bio-gene)+ antibiotics (20 mL/plate). Then add 5 mL of transformation mix per plate. Twenty-four hours after transfection, change

the culture media to DMEM + antibiotics without Serum; 72 hr after transfection, culture medium was collected and filtered to get rid of cell pellets. Collected medium was stirred at 4°C for 1.5 hr, meanwhile mixed with NaCl (final concentration of 0.4 M) and PEG8000 (final concentration of 8.5% w/v). Virus were precipitated by centrifugation at 7000 $g$ for 10 min. Supernatant was discarded and 10 mL lysis buffer (150 mM NaCl, 20 mM Tris pH = 8.0) was added to re-suspend the virus pellet. Virus was then concentrated and purified via iodixanol gradients ('Optiprep' Sigma D1556-250mL). Centrifuge the gradients for 90 min at 46,500 rpm at 16°C. The virus in 40% fraction was harvested and mixed with PBS and then transferred to an Amacon 100 K columns – UFC910008 to remove the iodixanol. Purity and titer of virus were then assessed by SDS-PAGE and SYPRO ruby staining (S-12000, Life Technologies, Carlsbad, CA).

## Viral and tracer injection

Mice were subjected to the surgical procedures described above. For viral injection into the EC, the following rostral parameters were used: anterior-posterior (AP) = 3.25 mm, medial-lateral (ML) = 3.80 mm, dorsal-ventral (DV) = 3.60 mm from the surface, volume = 100 nL. Similarly, the following caudal parameters were used: AP = 4.25 mm, ML = 3.60 mm, DV = 2.60 mm from surface, volume = 200 nL. For injection of tracer or virus into the LA, we used the following parameters: AP = 1.70 mm, ML = 3.40 mm, DV = 3.70 mm from the surface, volume = 200 nL. Craniotomy was performed after skull levelling and partial opening of the dura mater using a syringe needle hook (29 G). We used the Nanoliter2000 system (World Precision Instruments, Sarasota County, FL) for all infusions. Viral or tracer infusions were slowly pumped into brain tissue trough a fine-tip glass pipette filled with silicon oil at a speed of no more than 50 nL/min. After infusion, the pipette was left in the injection site for an extra 5–10 min before slow withdrawal. After withdrawal of the pipette, the scalp was sutured, and a local anesthetic was applied. The animal was returned to its home cage after awaking. For axon stimulation (observation), the virus was expressed for at least 7 weeks, and for cell body stimulation (observation), the virus was expressed for at least 4 weeks. For CTB tracer labeling, we perfused animals after 7 days of viral expression.

## Real-time PCR

To determine the expression of *Cck* after injecting our AAV carrying anti-*Cck* or anti-Scramble shRNA, real-time quantitative PCR (qPCR) was performed regarding the injection site (EC) and a reference site (contralateral AC). After expressing shRNA for at least 3 weeks, mice were deeply anesthetized with isoflurane (RWD, Shenzhen, China), and the brains were harvested. Tissue from target areas was collected and RNA from these tissues was first extracted by using Trizol (Cat# 15596018, Invitrogen, Waltham, MA) and then reverse-transcribed to cDNA with the PrimeScript RT Reagent Kit (Cat# RR037B, TaKaRA Bio Inc, Kusatsu, Shiga, Japan). Real-time PCR was performed by using TB Green Premix Ex Taq II (Cat# RR820A, TaKaRa). All samples were tested in triplicate. The primers used were listed in Table 1. The comparative cycle threshold (Ct) method (2^-ΔΔCt) was employed to calculate the relative level of gene expression. The housekeeping gene GAPDH was used to normalize the original Ct values in our current experiments.

## Optic fiber implantation

Mice were subjected to the surgical procedures described above. Craniotomy was performed bilaterally to target the LA using the coordinates described above. Optic fibers (optic cannulae) were gently inserted into the LA (50–100 μm above the target area) and fixed with dental cement (mega PRESS NV + JET X, megadental GmbH, Büdingen, Germany). For head fixation, a long screw was fixed to the skull with dental cement at a 45° angle from the vertical axis.

## Fiber photometry

The commercial 1-site Fiber Photometry System (Doric Lenses Inc, Quebec, Canada) coupled with the RZ5D processor (TDT, Alachua, FL) was used in the current study. Excitation light at 470 and 405 nm was emitted from two fiber-coupled LEDs (M470F3 and M405FP1, Thorlabs) and sinusoidally modulated at 210 and 330 Hz, respectively. The intensity of the excitation light was controlled by an LED driver (LEDD1B, Thorlabs) connected with the RZ5D processor via the software Synapse. Excitation light was delivered to the animal through a dichroic mirror embedded in single fluorescence MiniCube (Doric

Lenses, Quebec, QC, Canada) in a fiber-optic patch cord (200 µm, 0.37 NA, Inper, Hangzhou, China). The intensity of the excitation light at the tip of the patch cord was adjusted to less than 30 µW to avoid photobleaching. The emission fluorescence was collected and transmitted through a bandpass filtered by the MiniCube. The fluorescent signal was then detected, amplified, and converted to an analog signal by the photoreceiver (Doric Lenses). Finally, the analog signal was digitalized by the RZ5D processor and analyzed using Synapse software at 1 kHz with a 5 Hz low-pass filter.

Optical fiber implantation and fiber photometry were used to visualize CCK activity in vivo via a fluorescent sensor. Briefly, the GPCR activation-based CCK sensor (GRAB$_{CCK}$, AAV-hSyn-CCK2.0) was developed by inserting a circular-permutated green fluorescent protein (cpEGFP) into the intracellular domain of CCKBR (*Jing et al., 2019*). Binding of CCKBR with its endogenous or exogenous ligand (CCK) induces a conformational change in cpEGFP and results in increased fluorescence intensity, which we measured by fiber photometry.

## Chemogenetic manipulation

Each animal (with DREADD virus injection) received CLZ (0.5 mg/kg, Sigma-Aldrich, dissolved with 0.1% DMSO) or vehicle (sterilized saline with 0.1% DMSO) by intraperitoneal injection. After injection, animals were kept in transfer cages for 30 min to allow the drug to penetrate the BBB and bind to the DREADD receptor (*Gomez et al., 2017*). Animals were then placed in conditioning boxes for further training.

## Optogenetic manipulation

CCK-Cre mice were injected with AAV-EF1α-DIO-eNpHR3.0-mCherry or control AAV-hSyn-FLEX-GFP. After 6 weeks, animals received bilateral optic fiber implantation as described above. Mice were allowed a 1 week recovery before being subject to the long-trace or long-delay fear conditioning. Baseline freezing percentages were recorded in the test context on the pre-conditioning day as described above. On the conditioning day, mice were connected with the optic cables, which are relayed by a rotatory joint (Inper, Hangzhou, China) then connected to a 635 nm laser source (Inper, Hangzhou, China). For long-trace fear conditioning paradigm, the training procedures were described as above while the laser illumination was applied from the onset of the CS to the onset of the US with a frequency of 5 Hz (100 ms illumination +100 ms interval, 12 mW at tip). For long-delay fear conditioning paradigm, a 30-s-long CS was co-terminated with a 0.5-s-long US. Laser illumination with same intensity and frequency covered the whole CS presentation. Three trials of CS-US pairing were conducted in each training day and animals totally received 2 days of training. After training, on post-conditioning day, the conditioned response of the animal was recorded in the test context. All activity was captured by a camera on the ceiling and analyzed with the previously described Bonsai program.

## Anatomy and immunohistochemistry

Animals were anesthetized with an overdose of pentobarbital sodium, perfused with ice-cold phosphate buffered saline (PBS, 0.01 M, Sigma-Aldrich), and fixed with paraformaldehyde solution (PFA, 4% in PBS, Santa Cruz Biotechnology, Dallas, TX). Animals were decapitated, and the brain was gently removed and submerged into 4% PFA solution for additional fixation (~48 hr). Brains were sectioned into 40-µm-thick slices on vibratome (Leica VT1000 S). To observe viral expression, neural tracer labeling, or electrode track verification, sections were counter-stained with DAPI (1:10000, Santa Cruz Biotechnology) for 10 min and mounted onto slides with 70% glycerol (Santa Cruz Biotechnology) in PBS. For immunohistochemistry, sections were washed with 0.01 M PBS three times for 7 min each and blocked with blocking solution (5% goat serum and 0.1% Triton X-100 in PBS) at room temperature for 1.5 hr. Each primary antibody was diluted to the appropriate concentration (Table 1) in blocking solution and incubated on sections overnight at 4°C. The next day, sections were washed with PBS three times for 7 min each and stained with secondary antibody, which was prepared in PBST (0.1% Triton X-100 in PBS). Each secondary antibody was incubated on sections at room temperature for 3 hr. After secondary incubation, the sections were washed with PBS three times for 7 min each and counter-stained with DAPI for 10 min. Finally, sections were washed three times with PBS and mounted onto slides with 70% glycerol mounting medium. Fluorescent images were captured with a Nikon Eclipse Ni-E upright fluorescence microscope and a Zeiss LSM880 confocal microscope.

## Image analysis

Imaging signal analysis, including quantification of intensity and percent positivity, was conducted in Fiji (https://imagej.net/Fiji) (*Schindelin et al., 2012*). To quantify the number (percentage) of viral- or immunohistochemical-positive neurons, we used the Cell Counter plugin in Fiji. To quantify the projection intensity of viral-positive neural fibers, we used the FeatureJ plugin in Fiji. We applied Hessian filter to extract the fiber-like structures and converted the raw images to eigen images with smallest eigen values selected. Eigen images were then converted to binary image by applying a threshold in Fiji and pixel density was measured as the intensity of neural projection (*Grider et al., 2006*). To quantify the colocalization of the CCK+ terminal (CCK-EYFP and synaptophysin double positive) and the CCKBR-innervating CCK+ terminal (CCK-EYFP, synaptophysin, and CCKBR triple positive), we extracted the double-positive and triple-positive pixels in Fiji and adopted the pixel-based colocalization analysis algorithm from CellProfiler (https://cellprofiler.org/examples) (*McQuin et al., 2018*) to calculate the colocalization ratios.

## Statistical analysis

Group data are shown as mean ± SEM (standard error of the mean) unless otherwise stated. Statistical analyses, including two sample t-tests, paired sample t-tests, one-way RM ANOVA, and two-way RM ANOVA, were conducted in Origin 2018 (OriginLab, Northampton, MA) and SPSS 26 (IBM, Armonk, NY). Statistical significance was defined as $p < 0.05$ by default.

## Acknowledgements

Funding: The authors thank Eduardo Lau for administrative and technical assistance. This work was supported by Hong Kong Research Grants Council (T13-605/18W, 11102417M, 11101818M, 11103220), Natural Science Foundation of China (31671102), Health and Medical Research Fund (06172456 and 31571096), Innovation and Technology Fund (MRP/101/17X, MPF/053/18X, GHP_075_19GD). We also thank the following charitable foundations for their generous supports to JH: Wong Chun Hong Endowed Chair Professorship, Charlie Lee Charitable Foundation, and Fong Shu Fook Tong Foundation.

## Additional information

### Funding

| Funder | Grant reference number | Author |
| --- | --- | --- |
| University Grants Committee | T13-605/18-W 11102417M 11101818M 11103220 | Jufang He |
| Natural Science Foundation of China | 31671102 | Jufang He |
| Health and Medical Research Fund | 06172456 31571096 | Jufang He |
| Innovation and Technology Fund | MRP/101/17X MPF/053/18X GHP_075_19GD | Jufang He |

The funders had no role in study design, data collection and interpretation, or the decision to submit the work for publication.

### Author contributions

Hemin Feng, Data curation, Formal analysis, Investigation, Methodology, Project administration, Validation, Visualization, Writing – original draft; Junfeng Su, Methodology, Resources, Validation; Wei Fang, Investigation; Xi Chen, Methodology, Resources, Software; Jufang He, Conceptualization, Funding acquisition, Project administration, Resources, Supervision, Writing - review and editing

### Author ORCIDs

Hemin Feng  http://orcid.org/0000-0002-8390-6005

Xi Chen ⓘ http://orcid.org/0000-0002-2144-6584
Jufang He ⓘ http://orcid.org/0000-0002-4288-5957

### Ethics

All experimental procedures were approved by the Animal Subjects Ethics Sub-Committee of the City University of Hong Kong (Reference number of animal ethics review: A-0529 and A-0282).

### Decision letter and Author response

Decision letter https://doi.org/10.7554/eLife.69333.sa1
Author response https://doi.org/10.7554/eLife.69333.sa2

## Additional files

### Supplementary files

• Transparent reporting form

### Data availability

Data for this submission has been uploaded to the Dryad Digital Repository, doi:https://doi.org/10.5061/dryad.0p2ngf217.

The following dataset was generated:

| Author(s) | Year | Dataset title | Dataset URL | Database and Identifier |
|---|---|---|---|---|
| Feng H, Su J, Fang W, Chen X, He J | 2021 | The entorhinal cortex modulates trace fear memory formation and neuroplasticity in the lateral amygdala via cholecystokinin | https://doi.org/10.5061/dryad.0p2ngf217 | Dryad Digital Repository, 10.5061/dryad.0p2ngf217 |

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
