## [Editor Report]

While the amygdala is important for associating innocuous sensory stimuli with aversive outcomes during associative fear learning, the medial temporal lobe memory system, including the entorhinal cortex, participates in bridging temporal gaps (trace periods) between the sensory stimuli and aversive outcomes. However, the circuit connections between these structures that allow for trace fear learning have not been clarified. Here, Feng et al. reveal that a specific population of cholecystokinin cells in the entorhinal cortex that project to the lateral nucleus of the amygdala are important for trace fear memory formation.

---

## [Decision Letter]

**Decision letter after peer review:**

Thank you for submitting your article "The entorhinal cortex modulates trace fear memory formation and neuroplasticity in the lateral amygdala via cholecystokinin" for consideration by *eLife*. Your article has been reviewed by 3 peer reviewers, including Joshua Johansen as the Reviewing Editor and Reviewer #1, and the evaluation has been overseen by Kate Wassum as the Senior Editor. The following individuals involved in review of your submission have agreed to reveal their identity: Stephen Maren (Reviewer #2); Fabricio do Monte (Reviewer #3).

Essential revisions:

1) A central claim and potentially the most important scientific advance in the paper is the identification of a peptidergic entorhinal cortex (EC)-lateral amygdala (LA) pathway for trace fear conditioning. However, the authors have not shown definitively that this circuit is specifically involved in trace, as opposed to delay (overlapping CS and US) or context, fear conditioning. This is important as trace conditioning involves a distinct process, a trace period which needs to be bridged, which is not present in delay fear conditioning. This is particularly relevant for the manipulation experiments described in Figure 8 which most strongly support the existence of the EC-LA peptidergic pathway. Ideally, the authors would redo these optogenetic experiments and compare the effects of the optogenetic manipulation on both trace and delay conditioning using a more classical trace protocol (e.g. 10 sec CS and 20 sec trace period as in other parts of the paper) and possibly a long-delay control (https://pubmed.ncbi.nlm.nih.gov/16904919/) which does not have a trace period and instead extends the CS period to 30 seconds total to co-terminate with the shock US. If there is an effect of this manipulation on both trace and delay fear conditioning, then the authors would need to change the interpretation of their findings considerably.

2) In determining the effects of experimental manipulations on freezing scores, the primary behavioral readout in this study, the authors make inappropriate use of statistical tests. While the authors' comparisons of group averages for freezing are reasonable, the use of t-tests to compare the effects of manipulations across time during trials is inappropriate and would be better suited for repeated measures ANOVAs. This issue can be easily addressed by reanalysis of this set of data.

3) The use of shRNA knockdown in EC-CCK^+^ cells to reduce the optogenetic assisted enhancement of auditory processing plasticity in the LA (Figure 7) is an elegant test of whether CCK is the molecule in these cells which is responsible for the plasticity induction. However, the authors should also verify that their shRNA constructs are in fact targeting CCK. This could be done by quantifying CCK transcript/mRNA from EC punches and/or performing immunohistochemistry for CCK in virus treated experimental vs. control tissue.

Below are more detailed comments:

1) It looks from the WT example trace in Figure 2g that the CCK^-/-^ animals have larger amplitude and slope than the WT controls. Since the authors normalize to the baseline, it is possible that there is synaptic saturation in the CCK^-/-^ animals and that LTP is occluded, not that potentiation is blocked. The authors should examine this more thoroughly in some way, perhaps analyzing slope and amplitude before and after LTP induction in CCK^-/-^ and WT animals. This is also an issue for the data in Figure 6e-j and Figure 7 and should be addressed in a similar way.

2) For the AEP experiments, it is not clear what sound intensities were used and whether there were differences between groups. The authors say that they used sound intensities that were '50% of the maximum sound intensity at a 5 s intertrial interval…" for the baseline sessions and ~70% of the maximum sound intensity (defined as intensity that elicited saturated response) for the HFS experiments, but that is not enough information for the reader to understand what was done. The authors should explicitly describe what sound intensities were actually used for CCK^-/-^ and WT animals, possibly using a bar graph (to show averages) with individual animal data-points embedded.

3) The authors conclude that MGB doesn't contain LA projecting CCK^+^ cells based on their cre-dependent expression of retroAAV expressed fluorophore (Figure 3i). However, it is possible that retroAAV is not expressed in MGB cells because of tropism. The authors should inject a retroAAV-fluophore (no cre-dependent) into LA to determine whether the fluorophore is expressed in MGB to control for this possibility.

4) For the data in Figure 3k-l, the authors should quantify the proportion of retroAAV labeled cells which are CaMKII+ and GAD67+.

5) The use of shRNA knockdown in EC-CCK^+^ cells to reduce the optogenetic assisted enhancement of auditory processing plasticity in the LA (Figure 7) is a nice test of whether CCK is the molecule responsible for the plasticity induction. However, they should determine whether this approach is in fact cre-dependent by injecting the cre-dependent shRNA/ChR2 into the EC of WT mice. Most importantly, they should also verify that their shRNA constructs are in fact targeting CCK. They could do this by quantifying CCK transcript from EC punches and/or performing immuno for CCK in virus treated experimental vs. control tissue.

6) The authors previously demonstrated that CCK-deficient mice have impairments in delay fear conditioning (Chen et al., 2019). Hence the observed impairments in trace fear conditioning may reflect a general deficit in delay fear conditioning (rather than a specific deficit in trace conditioning). An important control to isolate the nature of deficits in animals undergoing trace fear conditioning is animals conditioned with the same CS-US inter stimulus interval – but without an intervening trace. These so-called "long-delay controls" de-confound the stimulus-free trace interval and the long CS-US ISI in the trace conditioning procedure.

7) For the final optogenetic experiment (Figure 8), the authors use a short trace period of 2 sec in contrast with the 20 sec trace period they use in most of the other experiments. This is an issue because prior work has shown that short trace periods like this are more similar to normal delay/traditional fear conditioning without a trace period and don't engage the hippocampus (Chowdhury et al. 2005). This is an important experiment because it provides the only direct, functional support for the idea that EC-CCK projections to LA regulate trace fear conditioning. The authors should redo this experiment with the longer 20 sec trace interval. They should also tone down their argument that this pathway is important specifically in trace fear conditioning as the short 2 sec trace experiment may in fact be a form of delay fear learning. If they want to explicitly test this then they should perform the same experiment with delay fear conditioning (i.e. overlapping CS and US).

8) Another concern is that it trace conditioning may depend, in part, on contextual conditioning – neural manipulations (eg, HPC lesions) that impair trace often cause contextual conditioning impairments. As the authors note, the EC and its projections to the amygdala have been implicated in contextual fear conditioning, so the observed impairments in trace might be due to underlying impairments on contextual fear conditioning. It would be important to understand whether CCK deficient mice have deficits in contextual conditioning or long-delay auditory fear conditioning. If so, there is nothing unique about the contribution of EC->LA projections to trace conditioning per se.

9) Statistical Considerations: For real-time freezing scores shown in several figures (Figure 1D and H, Figure 2O, Figure 4G and M), repeated measures ANOVAs would be more appropriate than t-tests for analyzing these data.

10) Methodological Considerations: For the experiments utilizing ChETA (Figure 6), it is important to note that the AAV9 serotype can be transported anterogradely, resulting in ChETA expressing in LA-CCK neurons. To address this possible issue, the authors should refer to this caveat in the text and, perhaps, show histology in LA with DAPI and EYFP overlayed to investigate the possibility of anterograde transport of the virus.

[Editors' note: further revisions were suggested prior to acceptance, as described below.]

Thank you for resubmitting your work entitled "The entorhinal cortex modulates trace fear memory formation and neuroplasticity in the mouse lateral amygdala via cholecystokinin" for further consideration by *eLife*. Your revised article has been reviewed by 3 peer reviewers, including Joshua Johansen as the Reviewing Editor and Reviewer #1, and the evaluation has been overseen by Kate Wassum as the Senior Editor. The following individuals involved in review of your submission have agreed to reveal their identity: Stephen Maren (Reviewer #2); Fabricio do Monte (Reviewer #3).

We agreed that the authors have done outstanding job addressing the many concerns raised in the earlier round of review. We think the long-delay controls and evidence that entorhinal cortical projections to the amygdala have some specificity for the trace conditioning procedure is especially important. The manuscript has been improved but there are a few remaining issues that need to be addressed to support the clarity and transparency of the manuscript, as outlined below:

– One small issue, on line 407-408 "We found that mice expressed Jaws (Exp, N = 8/3 cages) had a prominent higher freezing percentage than mice expressed mCherry control…" We think you meant to say the opposite, that Jaws expressing animals had lower freezing levels.

– One point that was raised in the earlier reviews, was that common mechanisms have been suggested to support trace and contextual fear conditioning, and that the authors may have identified a neural circuit that mediates both processes. This was addressed in the Discussion, but the likely possibility that EC-LA projections are also involved in contextual conditioning was not addressed. Please include.

– Clarify whether the quantification of freezing during the after-CS period for the short trace interval includes only the post-CS period (2 s) or also the duration of the CS (3 s). Provide the same information for the long trace interval experiments too.

– For Figure 6H, consider using a schematic similar to what you have used for Figure 3F. The current one is extremely hard to understand.

– Consider keeping the previous Figure 8 with the halorhodopsin inactivation of ECcck-LA pathway during short trace interval in the supplementary material rather than completely removing the results from the article.

– Please report exact p values for t-tests and ANOVAs, in line with *eLife*'s reporting requirements: https://reviewer.elifesciences.org/author-guide/full "Report exact p-values wherever possible alongside the summary statistics and 95% confidence intervals. These should be reported for all key questions and not only when the p-value is less than 0.05."

– Please include in your methods light cycle at test (presumably it is during the light cycle, but this wasn't clear in the first paragraph of the methods).

---

## [Author Response]

Essential revisions:1) A central claim and potentially the most important scientific advance in the paper is the identification of a peptidergic entorhinal cortex (EC)-lateral amygdala (LA) pathway for trace fear conditioning. However, the authors have not shown definitively that this circuit is specifically involved in trace, as opposed to delay (overlapping CS and US) or context, fear conditioning. This is important as trace conditioning involves a distinct process, a trace period which needs to be bridged, which is not present in delay fear conditioning. This is particularly relevant for the manipulation experiments described in Figure 8 which most strongly support the existence of the EC-LA peptidergic pathway. Ideally, the authors would redo these optogenetic experiments and compare the effects of the optogenetic manipulation on both trace and delay conditioning using a more classical trace protocol (e.g. 10 sec CS and 20 sec trace period as in other parts of the paper) and possibly a long-delay control (https://pubmed.ncbi.nlm.nih.gov/16904919/) which does not have a trace period and instead extends the CS period to 30 seconds total to co-terminate with the shock US. If there is an effect of this manipulation on both trace and delay fear conditioning, then the authors would need to change the interpretation of their findings considerably.

We appreciate your suggestion. We redid the inhibition effect of optogenetic manipulation in long-trace (10 s CS and 20 s trace interval) and long-delay (30 s CS co-terminated with 0.5 s US) experiments. The experimental group CCK-Cre mice that expressed AAV with the opto-inhibitory opsin in the EC had a significantly lower freezing percentage in the test session, compared to the control group CCK-Cre mice that expressed a control AAV virus in the EC, both after 16 training trials in the long-trace conditioning paradigm. However, in the long-delay conditioning paradigm, mice in the experimental group did not show an impaired freezing percentage compared to those in the control group. That means the optogenetic inhibition of the EC-LA CCK positive circuit can specifically interfere with the fear memory formation in long-trace conditioning, suggesting this circuit is specifically involved in trace rather than delay fear memory formation.

Please see the below quotes and Figure 8 in our revised main text for details:

Lines 367-400:

“We employed optogenetics to dissect the real-time behavioral dependency of the trace fear memory formation on the ECCCK+→LA pathway. […] From the freezing score plot on the test day (Figure 8f), we observed a similar response curve to the CS, with some time points, the experimental group had a higher freezing score than the control group (**P* < 0.05, two-way RM ANOVA with Bonferroni multiple pairwise comparison between two groups in each time point).”

“Collectively, with the real-time opto-inhibition on CCK projections from the EC to the LA, we found the specific involvement of the ECCCK+→LA in the trace fear memory formation.”

2) In determining the effects of experimental manipulations on freezing scores, the primary behavioral readout in this study, the authors make inappropriate use of statistical tests. While the authors' comparisons of group averages for freezing are reasonable, the use of t-tests to compare the effects of manipulations across time during trials is inappropriate and would be better suited for repeated measures ANOVAs. This issue can be easily addressed by reanalysis of this set of data.

We appreciate the comment. We re-analyzed the data with two-way RM ANOVA followed by the Bonferroni post-hoc multiple pairwise comparisons. All the relevant figure panels were updated accordingly.

3) The use of shRNA knockdown in EC-CCK^+^ cells to reduce the optogenetic assisted enhancement of auditory processing plasticity in the LA (Figure 7) is an elegant test of whether CCK is the molecule in these cells which is responsible for the plasticity induction. However, the authors should also verify that their shRNA constructs are in fact targeting CCK. This could be done by quantifying CCK transcript/mRNA from EC punches and/or performing immunohistochemistry for CCK in virus treated experimental vs. control tissue.

We agree with the reviewer this is an important issue and appreciate the suggestion. We injected the AAVs carrying anti-CCK shRNA and anti-scramble shRNA respectively into the EC of CCK-Cre mice. We dissected the EC (transfected area) and AC (untransfected area) from those mice. We performed the qPCR test to quantify the expression level of *Cck* mRNA. The expression level of *Cck* mRNA in the EC of the anti-scramble group was chosen as the normalization baseline (100 ± 19.8%). We found the *Cck* mRNA level in the EC of the anti-CCK group was significantly reduced (41.1% ± 4.7%). Meanwhile, the *Cck* mRNA level from the untransfected area did not have an observable difference (129.7% ± 16.2% in the AC from anti-scramble group vs. 131.0% ± 11.9% in the AC from anti-CCK group). In conclusion, our anti-CCK shRNA effectively knocked down the expression of *Cck*. We updated this result in panel D of Figure 7 in the revised version.

Below are more detailed comments:1) It looks from the WT example trace in Figure 2g that the CCK^-/-^ animals have larger amplitude and slope than the WT controls. Since the authors normalize to the baseline, it is possible that there is synaptic saturation in the CCK^-/-^ animals and that LTP is occluded, not that potentiation is blocked. The authors should examine this more thoroughly in some way, perhaps analyzing slope and amplitude before and after LTP induction in CCK^-/-^ and WT animals. This is also an issue for the data in Figure 6e-j and Figure 7 and should be addressed in a similar way.

Thanks for the comment. We have enough evidence to exclude the possibility of synaptic saturation leading to LTP occlusion. Firstly, based on the I-O curve we measured, we chose a moderate instead of the saturated intensity of noise stimulation in our baseline recording, which means we reserved the space of increase of AEP after LTP induction. Secondly, as the reviewer suggested, we analyzed the actual slope and amplitude of recorded fEPSP from the raw data. We found no significant difference between the slope in raw data or amplitude in the CCK^-/-^ and WT group, excluding the possibility that LTP blockage in CCK^-/-^ was caused by the ceiling effect of AEP. We added the data in panels D and E of Figure 2—figure supplement 1.

2) For the AEP experiments, it is not clear what sound intensities were used and whether there were differences between groups. The authors say that they used sound intensities that were '50% of the maximum sound intensity at a 5 s intertrial interval…" for the baseline sessions and ~70% of the maximum sound intensity (defined as intensity that elicited saturated response) for the HFS experiments, but that is not enough information for the reader to understand what was done. The authors should explicitly describe what sound intensities were actually used for CCK^-/-^ and WT animals, possibly using a bar graph (to show averages) with individual animal data-points embedded.

As suggested, we added a bar graph embedded with raw data to show the actual sound intensity was presented for each point. Data were added in panel (f) of Figure 2—figure supplement 1.

3) The authors conclude that MGB doesn't contain LA projecting CCK^+^ cells based on their cre-dependent expression of retroAAV expressed fluorophore (Figure 3i). However, it is possible that retroAAV is not expressed in MGB cells because of tropism. The authors should inject a retroAAV-fluophore (no cre-dependent) into LA to determine whether the fluorophore is expressed in MGB to control for this possibility.

Because we did not have retroAAV-fluophore, we used an alternative virus retroAAV-hSyn-Cre to confirm the tropism of retroAAV. We injected this retroAAV-Cre into the LA of Ai14 (Cre-dependent tdTomato reporter) mice (N=3). And we obtained highly similar results as to what from CTB tracer showed in Figure 3b-e. Besides the AC and EC, we also found retrograde tdTomato positive cell bodies at the MGB, which excluded the possible deficit that retroAAV was not expressed in MGB cells. The related images were updated in Figure S3. We also revised our interpretation that MGB has no CCK-positive projection to the LA in the revised main text as below:

“Considering the potential tropism of retroAAV that may cause the absence of AAV expression in the MGB, we injected a Cre-expressing retroAAV (retroAAV-hSyn-Cre) into the LA of the Cre-dependent tdTomato reporter Ai14 mice (N = 3). […] However, based on our ongoing studies, we cannot exclude the possible scenario that MGB may originate some CCK-positive projection to LA during some stages of development.”

4) For the data in Figure 3k-l, the authors should quantify the proportion of retroAAV labeled cells which are CaMKII+ and GAD67+.

We added the quantitative data of the proportion of CamKII+ and GAD67+ neurons in retroAAV labelled ones in Figure 3 in the revised version.

5) The use of shRNA knockdown in EC-CCK^+^ cells to reduce the optogenetic assisted enhancement of auditory processing plasticity in the LA (Figure 7) is a nice test of whether CCK is the molecule responsible for the plasticity induction. However, they should determine whether this approach is in fact cre-dependent by injecting the cre-dependent shRNA/ChR2 into the EC of WT mice. Most importantly, they should also verify that their shRNA constructs are in fact targeting CCK. They could do this by quantifying CCK transcript from EC punches and/or performing immuno for CCK in virus treated experimental vs. control tissue.

Thanks for your kind suggestion. We injected this Cre-dependent shRNA/ChR2 into the EC of WT mice and confirmed the Cre-dependency of the virus as immunostaining detected no ChR2 signal in the slices from these WT mice. This result was updated in the Figure 7—figure supplement 1. Meanwhile, we also used qPCR to verify the knockdown efficiency of our shRNA in CCK-Cre mice. The detailed answer about qPCR verification is shown in our response to Essential Revision 3.

We also revised our main text accordingly:

Line 354-357

“The knock-down efficiency on *Cck* expression was quantitively verified by real-time PCR (Figure 7d). Meanwhile, we injected this virus in WT mice and found ChR2 was not expressed in the injected area, indicating a reliable Cre-dependency of this AAV (Figure 7—figure supplement 1).”

6) The authors previously demonstrated that CCK-deficient mice have impairments in delay fear conditioning (Chen et al., 2019). Hence the observed impairments in trace fear conditioning may reflect a general deficit in delay fear conditioning (rather than a specific deficit in trace conditioning). An important control to isolate the nature of deficits in animals undergoing trace fear conditioning is animals conditioned with the same CS-US inter stimulus interval – but without an intervening trace. These so-called "long-delay controls" de-confound the stimulus-free trace interval and the long CS-US ISI in the trace conditioning procedure.

We appreciate the comments and the suggestion. We redid our opto-inhibition experiment to clarify the specific modulation effect of the EC-LA CCK positive pathway on long trace conditioning. As the reviewer suggested, we used the long-delay control to segregate the deficit of animals in trace fear conditioning. We found this pathway is involved explicitly in trace conditioning. However, the deficiency in CCK-KO mice may be broader than optogenetic inhibition on the EC-LA CCK pathway, as stated in the revised Discussion.

“We previously reported that WT animals form CS-US associations after three training trials with minimal fear generalization in auditory-cued delay fear conditioning. […] Together, the results of our previous work and the present study indicate that the absence of the neuropeptide CCK has broad damaging effects on multiple forms of fear memory and is not limited to trace fear memory.”

7) For the final optogenetic experiment (Figure 8), the authors use a short trace period of 2 sec in contrast with the 20 sec trace period they use in most of the other experiments. This is an issue because prior work has shown that short trace periods like this are more similar to normal delay/traditional fear conditioning without a trace period and don't engage the hippocampus (Chowdhury et al. 2005). This is an important experiment because it provides the only direct, functional support for the idea that EC-CCK projections to LA regulate trace fear conditioning. The authors should redo this experiment with the longer 20 sec trace interval. They should also tone down their argument that this pathway is important specifically in trace fear conditioning as the short 2 sec trace experiment may in fact be a form of delay fear learning. If they want to explicitly test this then they should perform the same experiment with delay fear conditioning (i.e. overlapping CS and US).

Thanks for the suggestion. We redid our opto-inhibition experiment. We found the intervening of the EC-LA CCK pathway during fear conditioning training impaired the performance of mice in the long-trace paradigm while leaving that in the long-delay paradigm unaffected. Please refer to our answer to Essential Revision 1 for details.

8) Another concern is that it trace conditioning may depend, in part, on contextual conditioning – neural manipulations (eg, HPC lesions) that impair trace often cause contextual conditioning impairments. As the authors note, the EC and its projections to the amygdala have been implicated in contextual fear conditioning, so the observed impairments in trace might be due to underlying impairments on contextual fear conditioning. It would be important to understand whether CCK deficient mice have deficits in contextual conditioning or long-delay auditory fear conditioning. If so, there is nothing unique about the contribution of EC->LA projections to trace conditioning per se.

Thanks for the comments. As added in the revised Discussion, the lack of CCK can bring general impairment to both trace and (short) delay fear conditioning:

“We previously reported that WT animals form CS-US associations after three training trials with minimal fear generalization in auditory-cued delay fear conditioning. […] Together, the results of our previous work and the present study indicate that the absence of the neuropeptide CCK has broad damaging effects on multiple forms of fear memory and is not limited to trace fear memory.”

Also, the lesion of the EC impairs the trace conditioning while leaves the delay conditioning unaffected (Esclassan et al., 2009, JNS). Our other ongoing studies found deficits in spatial memory formation in the Morris water maze test in CCK-KO mice. Therefore, we cannot guarantee the uniqueness of the deficiency in CCK-KO mice. Our current study focused on clarifying the function of the EC-LA CCK-positive pathway. Following the suggestions from reviewers, we conducted the opto-inhibition experiment on both long-trace and long-delay conditioning, and we found the uniqueness of this pathway in the formation of long-trace fear memory instead of long-delay fear memory.

9) Statistical Considerations: For real-time freezing scores shown in several figures (Figure 1D and H, Figure 2O, Figure 4G and M), repeated measures ANOVAs would be more appropriate than t-tests for analyzing these data.

Thanks for the suggestion, and we re-analyzed the relevant data with two-way RM ANOVA followed by Bonferroni multiple pairwise comparisons.

10) Methodological Considerations: For the experiments utilizing ChETA (Figure 6), it is important to note that the AAV9 serotype can be transported anterogradely, resulting in ChETA expressing in LA-CCK neurons. To address this possible issue, the authors should refer to this caveat in the text and, perhaps, show histology in LA with DAPI and EYFP overlayed to investigate the possibility of anterograde transport of the virus.

Thanks for this kind suggestion. After re-examination of the histological images showing the overlay of DAPI and EYFP at LA, we could exclude the post-synaptic labelling of LA-CCK neurons caused by AAV9 anterograde transportation. We could hardly observe the EYFP positive cell bodies at LA. We could think of the following two possibilities. The virus we used in Figure 6 (AAV9-EF1a-DIO-CHETA-EYFP) uses EF1a as the promoter, which could be less potent than the reported hSyn and CMV promoter (Zingg et al., Neuron, 2016) in anterograde transmission. Another possibility is that the viral titer we used (around 5e12 GC/ml) is much lower than reported in the above paper (more than 1e13 GC/ml). We updated the related images in Figure 6 in the new version.

[Editors' note: further revisions were suggested prior to acceptance, as described below.]

We agreed that the authors have done outstanding job addressing the many concerns raised in the earlier round of review. We think the long-delay controls and evidence that entorhinal cortical projections to the amygdala have some specificity for the trace conditioning procedure is especially important. The manuscript has been improved but there are a few remaining issues that need to be addressed to support the clarity and transparency of the manuscript, as outlined below:– One small issue, on line 407-408 "We found that mice expressed Jaws (Exp, N = 8/3 cages) had a prominent higher freezing percentage than mice expressed mCherry control…" We think you meant to say the opposite, that Jaws expressing animals had lower freezing levels.

Thanks for correcting our mistake, and we revised our text as follows:

“Freezing percentage to the CS was measured before (baseline) and after (post-training) this long trace fear conditioning (Figure 8e). We found that mice expressed Jaws (Exp, N = 8/3 cages) had a prominent lower freezing percentage than mice expressed mCherry control (Ctrl, N = 9/3 cages), while in baseline session, there is no statistical difference between these groups (Figure 8e, two-way RM ANOVA, significant interaction, F[1,15] = 5.59, *P* = 0.032 < 0.05; in baseline session, Exp vs. Ctrl, 7.8 ± 2.1% vs. 11.6 ± 2.0%; 95% CI, [3.4%, 12.2%] vs. [7.4%, 15.7%]; *P* = 0.208 > 0.05; in post-training session, Exp vs. Ctrl, 33.3 ± 5.3 % vs. 51.9 ± 5.0 %; 95% CI, [22.1%, 44.5%] vs. [41.4%, 62.5%]; *P* = 0.021 < 0.05; Video 11-12).”

– One point that was raised in the earlier reviews, was that common mechanisms have been suggested to support trace and contextual fear conditioning, and that the authors may have identified a neural circuit that mediates both processes. This was addressed in the Discussion, but the likely possibility that EC-LA projections are also involved in contextual conditioning was not addressed. Please include.

Thanks for the suggestion. We included the following paragraph in our Discussion.

“Of note, we cannot underestimate the dependence of trace fear memory on contextual fear memory because some critical areas, include the hippocampus (McEchron et al., 1998) and the medial prefrontal cortex (Gilmartin and Helmstetter, 2010), contribute to both types of fear memory. […] Therefore, our unveiled CCK positive EC-LA projections may also involve the formation of contextual fear.”

– Clarify whether the quantification of freezing during the after-CS period for the short trace interval includes only the post-CS period (2 s) or also the duration of the CS (3 s). Provide the same information for the long trace interval experiments too.

Thanks for this suggestion, and we added the following description in our main text:

"We calculated the freezing percentage during the after-CS period including the duration of CS (10 s) and the trace interval (20 s) and the before-CS period with the time window of the same length of 30 s immediately before the CS presentation.”

"Same as above, we calculated the freezing percentage during the after-CS period including the duration of CS (3 s) and the trace interval (2 s) and during the before-CS period with the same time window of 5 s immediately before the CS presentation."

– For Figure 6H, consider using a schematic similar to what you have used for Figure 3F. The current one is extremely hard to understand.

Thanks for this suggestion. We revised our Figure 6H.

– Consider keeping the previous Figure 8 with the halorhodopsin inactivation of ECcck-LA pathway during short trace interval in the supplementary material rather than completely removing the results from the article.

Thanks for the suggestion, and we put it back as our figure supplement to current figure 8.

– Please report exact p values for t-tests and ANOVAs, in line with eLife's reporting requirements: https://reviewer.elifesciences.org/author-guide/full "Report exact p-values wherever possible alongside the summary statistics and 95% confidence intervals. These should be reported for all key questions and not only when the p-value is less than 0.05."

Thanks for the suggestion, and we report the exact p-value and 95% confidence interval in our updated text. Two examples are listed below.

“At baseline, *Cck*^-/-^ (N = 10/2 cages) and WT (N = 14/3 cages) mice showed similarly low freezing percentages both before (Figure 1b) and after (Figure 1c) the CS (Figure 1b, two-way repeated-measures analysis of variance [RM ANOVA], significant interaction, F [1,22] = 10.85, *P* = 0.003 < 0.01; pairwise comparison, WT vs. *Cck*^-/-^ before CS, 7.0% ± 1.0% vs. 5.9% ± 1.1%; 95% confidence interval [CI], [5.0%, 9.0%] vs. [3.6%, 8.3%]; Bonferroni test, *P* = 0.482 > 0.05; Figure 1c, two-way RM ANOVA, significant interaction, F [1,22] = 8.94, *P* = 0.007 < 0.01; pairwise comparison, WT vs. *Cck*^-/-^ after CS, 9.9% ± 1.5% vs. 9.6% ± 1.8%; 95% CI, [6.8% 13.0%] vs. [5.9% 13.3%]; Bonferroni test, *P* = 0.911 > 0.05).”

“We found that mice expressed Jaws (Exp, N = 8/3 cages) had a prominent lower freezing percentage than mice expressed mCherry control (Ctrl, N = 9/3 cages), while in baseline session, there is no statistical difference between these groups (Figure 8e, two-way RM ANOVA, significant interaction, F[1,15] = 5.59, *P* = 0.032 < 0.05; in baseline session, Exp vs. Ctrl, 7.8 ± 2.1% vs. 11.6 ± 2.0%; 95% CI, [3.4%, 12.2%] vs. [7.4%, 15.7%]; *P* = 0.208 > 0.05; in post-training session, Exp vs. Ctrl, 33.3 ± 5.3 % vs. 51.9 ± 5.0 %; 95% CI, [22.1%, 44.5%] vs. [41.4%, 62.5%]; *P* = 0.021 < 0.05; Video 11-12).”

– Please include in your methods light cycle at test (presumably it is during the light cycle, but this wasn't clear in the first paragraph of the methods).

Thanks for the suggestion. We housed our animals in a light/dark reverse room and performed our behavioral experiments in the animals' dark cycle. Please see our revised text below.

"Mice were housed in a 12-hour shift of the reversed light-dark cycle and were provided food and water ad libitum. All behavioral experiments were conducted in the dark cycle."